# Neuropeptide F regulates courtship in *Drosophila* through a male-specific neuronal circuit

**Weiwei Liu[1†‡], Anindya Ganguly[1†], Jia Huang[2], Yijin Wang[1], Jinfei D Ni[1], Adishthi S Gurav[1], Morris A Aguilar[1], Craig Montell[1]***

[1]Department of Molecular, Cellular and Developmental Biology, and the Neuroscience Research Institute, University of California, Santa Barbara, Santa Barbara, United States; [2]Institute of Insect Sciences, Zhejiang University, Hangzhou, China

**Abstract** Male courtship is provoked by perception of a potential mate. In addition, the likelihood and intensity of courtship are influenced by recent mating experience, which affects sexual drive. Using *Drosophila melanogaster*, we found that the homolog of mammalian neuropeptide Y, neuropeptide F (NPF), and a cluster of male-specific NPF (NPF$^M$) neurons, regulate courtship through affecting courtship drive. Disrupting NPF signaling produces sexually hyperactive males, which are resistant to sexual satiation, and whose courtship is triggered by sub-optimal stimuli. We found that NPF$^M$ neurons make synaptic connections with P1 neurons, which comprise the courtship decision center. Activation of P1 neurons elevates NPF$^M$ neuronal activity, which then act through NPF receptor neurons to suppress male courtship, and maintain the proper level of male courtship drive.
DOI: https://doi.org/10.7554/eLife.49574.001

**\*For correspondence:**
cmontell@ucsb.edu

[†]These authors contributed equally to this work

**Present address:** [‡]Kunming Institute of Zoology, Chinese Academy of Sciences, Kunming, China

**Competing interests:** The authors declare that no competing interests exist.

## Introduction

To mate is a critical decision that sexually reproductive animals must make to ensure propagation of their species. Although courtship is an innate behavior that can be evoked mostly by stimulatory cues emitted from a conspecific of the opposite sex, the intensity of courtship is largely under control of animal's internal drive state, which is dictated in part by sexual satiation or deprivation.

The fruit fly, *Drosophila melanogaster*, has served as an animal model to decipher the genetic and neural basis of courtship behavior (*Sturtevant, 1915*; *Hotta and Benzer, 1976*; *Dickson, 2008*; *Villella and Hall, 2008*; *Robinett et al., 2010*; *Pavlou and Goodwin, 2013*; *Yamamoto and Koganezawa, 2013*). Distinct male and female courtship rituals (*Dickson, 2008*) are orchestrated by sexually dimorphic neuronal circuits, which are specified by the master transcriptional regulators – Fruitless and Doublesex (*Burtis and Baker, 1989*; *Ito et al., 1996*; *Ryner et al., 1996*; *Lee et al., 2000*; *Demir and Dickson, 2005*; *Manoli et al., 2005*; *Stockinger et al., 2005*; *Rideout et al., 2010*).

A male-specific cluster of P1 neurons, comprise the courtship decision center, which integrates multi-modal sensory inputs, and sends outputs to command neurons directing courtship behavior (*Kimura et al., 2008*; *Yu et al., 2010*; *Kohatsu et al., 2011*; *von Philipsborn et al., 2011*; *Pan et al., 2012*; *Bath et al., 2014*; *Inagaki et al., 2014*; *Clowney et al., 2015*; *Kallman et al., 2015*; *Kohatsu and Yamamoto, 2015*; *Zhou et al., 2015*). Besides integrating external stimuli, P1 neuronal activity is also influenced by the male's internal drive state, in part through dopaminergic neurons (*Zhang et al., 2016*).

Male flies exhibit escalated levels of courtship following periods of sexual deprivation. Conversely, courtship decreases once males become sexually satiated, following mating with an abundance of female partners (*Zhang et al., 2016*). Sexual deprivation induces higher excitability of P1 neurons, while excitability of P1 neurons is down-regulated in sexually satiated flies (*Inagaki et al., 2014*). Proper activation of P1 neurons allows male flies to display courtship when external sensory cues from potential mates match with their internal drive states. However, we have a poor understanding concerning the identities of the neuromodulators and associated neurons that impact on the courtship decision center, which responds to mating experience.

Neuropeptide F (NPF) (*Brown et al., 1999*) is a neuromodulator, and is a candidate for fine-tuning courtship by the internal state since NPF neural circuitry is sexually dimorphic (*Lee et al., 2006*; *Kim et al., 2013*), and because NPF expression levels and intracellular $Ca^{2+}$ activity in NPF neurons are altered by the animal's mating status (*Shohat-Ophir et al., 2012*; *Gao et al., 2015*). However, it is not known whether NPF is essential for male courtship, and the subsets of NPF neurons critical for courtship regulation have not been identified. Moreover, the neurons that interact with the sexually dimorphic NPF neurons to regulate courtship have not been defined.

Here, we show that a cluster of male-specific NPF neurons (NPF$^M$) are essential for regulating male courtship. Disrupting NPF signaling, either by knocking out *npf*, or by suppressing the activity of NPF neurons, reduces inhibition on courtship behavior in sexually satiated males, and evokes hypersexual activity in deprived males towards inappropriate targets. By combining anatomical, chemogenetic manipulation and $Ca^{2+}$ imaging approaches, we found that P1 neurons directly activate NPF$^M$ neurons, which then act through NPF receptor (NPFR) neurons to suppress male courtship. Our findings indicate that NPF signaling impinges on a dedicated male circuit and is critical for fine-tuning male courtship, in accordance with the internal sexual drive state.

## Results

### Disruption of NPF signaling elevates male courtship

To explore the potential function of NPF in modulating male sex drive, we first inhibited NPF neurons, and assayed its effects on male-female (M–F) and male-male (M–M) courtship. To inactivate NPF neurons, we employed three approaches in conjunction with the *Gal4/UAS* binary system (*Brand and Perrimon, 1993*). First, we used flies expressing Shibire$^{ts}$ (Shi$^{ts}$),which prevents synaptic transmission above 28 °C, due to depletion of synaptic vesicles (*Grigliatti et al., 1973*; *Poodry and Edgar, 1979*; *van der Bliek and Meyerowitz, 1991*; *Kuromi and Kidokoro, 1998*). We conducted the analysis using males that were group-housed with females, and experienced courtship and mating. Consequently, the males were sexually satiated and had lower courtship drive. When we disrupted synaptic transmission from NPF neurons, a higher percentage of males initiated courtship towards female targets, and their courtship index (ratio of time displaying courtship) was elevated significantly (*Figure 1A and B*). Courtship was not increased simply by raising the temperature, because flies expressing only the *npf-Gal4* or the *UAS-Shi$^{ts}$* exhibited similar courtship at both 23°C and 31°C (*Figure 1A and B*). At the restrictive temperature (31°C), *npf-Gal4/+;UAS-Shi$^{ts}$/+* males also vigorously engaged in courting other males and frequently formed courtship chains in which multiple males simultaneously court the preceding male (*Huang et al., 2016*) (*Figure 1C*). In contrast, we rarely detected chaining events among males kept at the permissive temperature (23°C; *Figure 1C*). We observed similar increases in M–M courtship when we inhibited the activity of NPF neurons by expressing a gene encoding either a hyperpolarizing $K^+$ channel (*Kir2.1*) (*Baines et al., 2001*) or the diphtheria toxin gene, *DTI* (*Han et al., 2000*) (*Figure 1—figure supplement 1A*). Inhibition of NPF neurons with Kir2.1 or DTI also eliminated M–M lunges, indicating suppression of aggression (*Figure 1—figure supplement 1B*). Conversely, when we increased NPF activity by constitutively expressing a $Na^+$ channel gene (*NaChBac*) (*Nitabach et al., 2006*) or by overexpressing the *npf-cDNA* (*Wu et al., 2003*) in NPF neurons, tester males exhibited elevated aggression (*Figure 1—figure supplement 1B*), but very few M–M courtship events (*Figure 1—figure supplement 1A*).

To clarify if it is the molecule NPF, rather than just NPF neurons, which is responsible for regulating sex drive, we generated two *npf* null mutants. To create *npf$^{LexA}$*, we replaced the *npf* gene with the *LexA* reporter using CRISPR-HDR (*Figure 1D* and *Figure 1—figure supplement 2A,B and C*) .

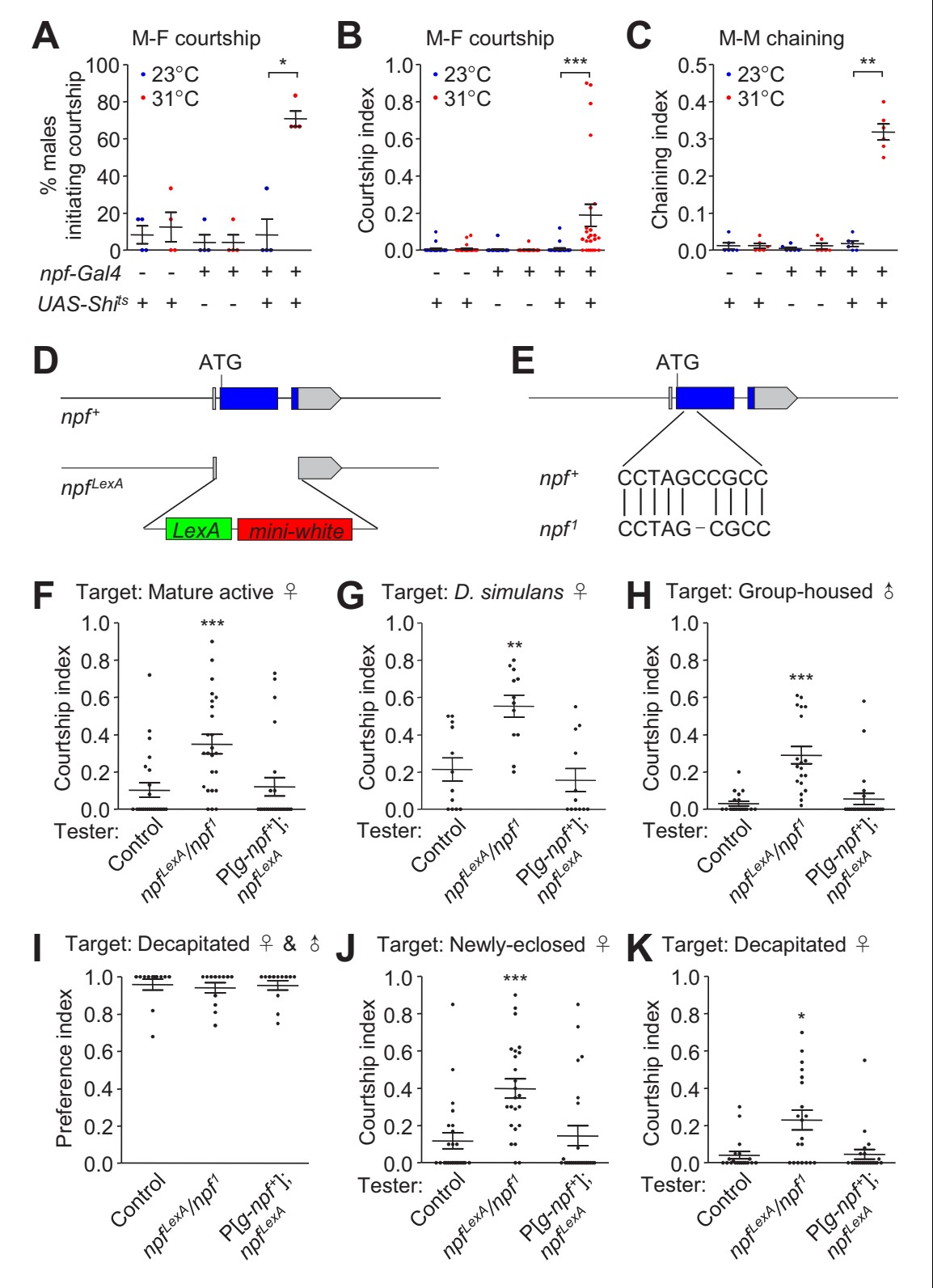

**Figure 1.** Effects of disruption of NPF neurons and the *npf* gene on male courtship. (**A and B**) Effects of silencing NPF neurons with Shi[ts] (*npf-Gal4/+*; *UAS-Shi[ts]/+*) on courtship of group-housed male flies towards female targets. Male-female (M–F) courtship was assayed at the permissive (23°C) and non-permissive (31 °C) temperatures for Shi[ts]. (**A**) The percentages of males that initiated courtship. n = 4 (6 flies/group). (**B**) Courtship index (ratio of time that a male fly exhibits courtship behavior out of the total observation time) scored from 20 to 30 min observation time during a 30 min incubation
*Figure 1 continued on next page*

*Figure 1 continued*

period, n = 24. (C) Silencing NPF neurons with Shi<sup>ts</sup> (*npf-Gal4/+;UAS-Shi<sup>ts</sup>/+*) induces male-male (M–M) courtship. Isolation-housed males were assayed for chaining behavior at 23°C and 31°C for 10 min. n = 6 (8—12 flies/group). The chaining index is the proportion of time that ≥ 3 tester males engage in courtship simultaneously out of a 10 min observation time. The bars indicate means ± SEMs. To determine significance, we used the Mann-Whitney test. *p < 0.05, **p < 0.01, ***p < 0.001. (D) Schematic illustration of *npf<sup>LexA</sup>* knock-in reporter line generated by the CRISPR-HDR method. (E) Schematic illustration of the *npf<sup>1</sup>* allele generated by the CRISPR-NHEJ method. *npf<sup>1</sup>* harbors a single nucleotide deletion in the 2<sup>nd</sup> position of codon 19. (F) Courtship index of group-housed males towards mature, active females. The control flies are *w<sup>1118-CS</sup>*. P[*g-npf<sup>+</sup>*] is a transgene encompassing the *npf<sup>+</sup>* genomic region. n = 24. (G) Courtship of isolation-housed males towards *Drosophila simulans* females. n = 12. (H) Courtship of isolation-housed males towards group-housed *w<sup>1118</sup>* males. n = 19—24. (I) Discrimination of male and female targets by the indicated males. Males of the indicated genotypes were exposed to a decapitated 5 day old male and a decapitated 5 day old virgin female. The preference index indicates the proportion of courtship time directed towards a female target out of the total courtship time in 10 min. A preference index of 1.0 indicates that the male spent 100 % of the time courting the decapitated female. n = 12. (J) Courtship index of group-housed males towards newly-eclosed female targets. n = 24. (K) Courtship of group-housed males towards decapitated female targets. n = 20—22. The bars indicate means ± SEMs. The Kruskal-Wallis test followed by the Dunn's *post hoc* test was used to assess significance. *p < 0.05, **p < 0.01, ***p < 0.001.
DOI: https://doi.org/10.7554/eLife.49574.002

The following source data and figure supplements are available for figure 1:

**Source data 1.** *Figure 1A—C* Source data.
DOI: https://doi.org/10.7554/eLife.49574.014
**Source data 2.** *Figure 1A—C* Summary statistics.
DOI: https://doi.org/10.7554/eLife.49574.015
**Source data 3.** *Figure 1F—K* Source data.
DOI: https://doi.org/10.7554/eLife.49574.016
**Source data 4.** *Figure 1F—K* Summary statistics.
DOI: https://doi.org/10.7554/eLife.49574.017
**Figure supplement 1.** Effects of increasing or decreasing NPF signaling on male-male (M–M) courtship and aggression.
DOI: https://doi.org/10.7554/eLife.49574.003
**Figure supplement 1—source data 1.** *Figure 1—figure supplement 1A* Source data.
DOI: https://doi.org/10.7554/eLife.49574.004
**Figure supplement 1—source data 2.** *Figure 1—figure supplement 1A* Summary statistics.
DOI: https://doi.org/10.7554/eLife.49574.005
**Figure supplement 1—source data 3.** *Figure 1—figure supplement 1B* Source data.
DOI: https://doi.org/10.7554/eLife.49574.006
**Figure supplement 1—source data 4.** *Figure 1—figure supplement 1B* Summary statistics.
DOI: https://doi.org/10.7554/eLife.49574.007
**Figure supplement 2.** Genotyping, testing for NPF expression, and courtship assays using the *npf<sup>1</sup>* and *npf<sup>LexA</sup>* mutants.
DOI: https://doi.org/10.7554/eLife.49574.008
**Figure supplement 2—source data 1.** *Figure 1—figure supplement 2F* Source data.
DOI: https://doi.org/10.7554/eLife.49574.009
**Figure supplement 2—source data 2.** *Figure 1—figure supplement 2F* Source data.
DOI: https://doi.org/10.7554/eLife.49574.010
**Figure supplement 2—source data 3.** *Figure 1—figure supplement 2G* Source data.
DOI: https://doi.org/10.7554/eLife.49574.011
**Figure supplement 2—source data 4.** *Figure 1—figure supplement 2G* Summary statistics.
DOI: https://doi.org/10.7554/eLife.49574.012
**Figure supplement 3.** Illustration of the aggression chamber.
DOI: https://doi.org/10.7554/eLife.49574.013

We also used the CRISPR-NHEJ method to generate an allele with a single nucleotide deletion, thereby changing the reading frame within codon 19, resulting in a null *npf* allele (*npf<sup>1</sup>*; *Figure 1E* and *Figure 1—figure supplement 2B and E*).

The courtship index of isolated control males reaches a ceiling level when they were exposed to mature active female targets (*Huang et al., 2016*). Therefore, to test whether *npf* mutant males exhibit an increase in courtship, we used mixed sex group-housed males, which in control flies showed a moderate level of courtship activity due to sexual satiation in the presence of an abundance of females. We found that mixed sex group-housed *npf* mutant males (*npf<sup>LexA</sup>* and *npf<sup>1</sup>* and the trans-heterozygous *npf<sup>LexA</sup>/npf<sup>1</sup>*) retained high levels of courtship towards mature active female

targets (*Figure 1F* and *Figure 1—figure supplement 2F*), indicating the mutants were resistant to sexual satiety induced by group-housing.

We found that the hypersexual activity in *npf* mutant males was generalized towards normally undesirable targets. These include females of other *Drosophila* species such as *D. simulans* females (*Clowney et al., 2015*) (*Figure 1G*), and male target flies (*Figure 1H* and *Figure 1—figure supplement 2G*). The increased courtship towards males was not due to an inability to discriminate between males and females. When the mutant males were allowed to choose between a decapitated male and a decapitated female, they showed a strong preference for female targets, similar to control males (*Figure 1I*).

To determine if this increase in courtship is an outcome of sensitized pheromone detection, we introduced newly-eclosed females, which carry negligible cuticular hydrocarbons and are therefore odorless/tasteless targets to tester males (*Liu et al., 2011*). We found that compared to control males, *npf* mutant males (*npf^LexA^/npf^1^*) exhibited significantly higher levels of courtship towards these females (*Figure 1J*). Thus, elevated courtship exhibited by *npf* mutant males did not appear to be caused by sensitized perception of attractive female pheromones. To test the possibility that the higher courtship levels was due to higher visual alertness in the mutants, we combined the males with motionless decapitated females as targets. Compared to control males, *npf* mutants exhibited increased courtship towards decapitated females (*Figure 1K*). To further establish that the courtship phenotype was due to loss of *npf*, we performed phenotypic rescue experiments with a wild-type *npf* genomic transgene (P[*g-npf^+^*], which restored *npf* expression to the *npf* mutant (*Figure 1—figure supplement 2B—D*). This genomic transgene also rescued normal levels of male courtship behavior to the *npf* mutant males (*Figure 1F—H,J and K*). Together, these experiments indicate that loss of *npf* function stimulates a sexually hyperactive state in males.

## Sexually dimorphic NPF^M^ neurons suppress male courtship

To examine the spatial distribution of the NPF neurons, we expressed *lexAop-IVS-mVenus* under the control of the *LexA* that we knocked into the *npf* gene (*npf^LexA^/+*). Among the neurons that were labeled by the *npf* reporter, was a bilaterally symmetrical cluster of NPF neurons that was male specific (NPF^M^; *Figure 2A and B*). The cell bodies of these sexually-dimorphic neurons are dorso-lateral to the antennal lobes and arborize extensively in the superior brain (*Figure 2A and B*). NPF^M^ neurons can be differentiated from other NPF neurons based on their position in the anterior brain region that is immediately adjacent to antennal lobe. Moreover, NPF^M^ form a cluster of 3— 5 neurons and their cell bodies are smaller than the pair of dorsal medial and the pair of dorsal lateral large NPF neurons. We used anti-NPF antibodies to immunostain the brain and found that the reporter expression pattern recapitulates the spatial distribution of the NPF protein (*Figure 2C, c1-c6*), confirming that NPF^M^ neurons express NPF.

The *fruitless* (*fru*) gene is a master regulator of male courtship behavior, and its role is mediated through expression of a male-specific protein, FruM, which is produced through alternative mRNA splicing (*Ito et al., 1996*; *Ryner et al., 1996*; *Lee et al., 2000*; *Demir and Dickson, 2005*; *Manoli et al., 2005*; *Stockinger et al., 2005*). We examined whether NPF^M^ neurons expressed the FruM protein by performing double labeling using anti-FruM, and anti-GFP, which marks the cells expressing *mVenus* driven by the *npf^LexA^* reporter. We found that NPF^M^ neurons expressed FruM (*Figure 2D, d1—d6*), while they were negative for DsxM (*Figure 2—figure supplement 1*), another key protein regulating the organization of the male nervous system (*Rideout et al., 2010*; *Robinett et al., 2010*).

To determine if FruM is essential for determining the fate of NPF^M^ neurons, we used anti-NPF antibodies to stain the brains of *fru^FLP^* mutant (*Yu et al., 2010*) males. We found that staining of NPF^M^ neurons was eliminated in *fru^FLP^* males (*Figure 2E and F*), indicating that specification of NPF^M^ neurons depends on FruM.

To distinguish the projection pattern of NPF^M^ neurons from the remaining NPF neurons, we used the FlpOut method (*Wong et al., 2002*) to specifically label NPF^M^ neurons with mCitrine. In the absence of both *fru^FLP^* and *npf^LexA^*, neither *mCherry* nor *mCitrine* is expressed (*Figure 3A*). If the flies contain the *fru^FLP^* transgene but not the *npf^LexA^* transgene, the *mCherry* gene is removed due to expression of Flp (FlpOut), but *mCitrine* is not expressed (*Figure 3B*). In flies with *npf^LexA^* but no *fru^FLP^*, *mCherry* is expressed, but *mCitrine* is not expressed due to the transcriptional *stop* cassette downstream of the coding region for *mCherry* (*Figure 3C*). If flies harbor both the *fru^FLP^* and *npf^LexA^*

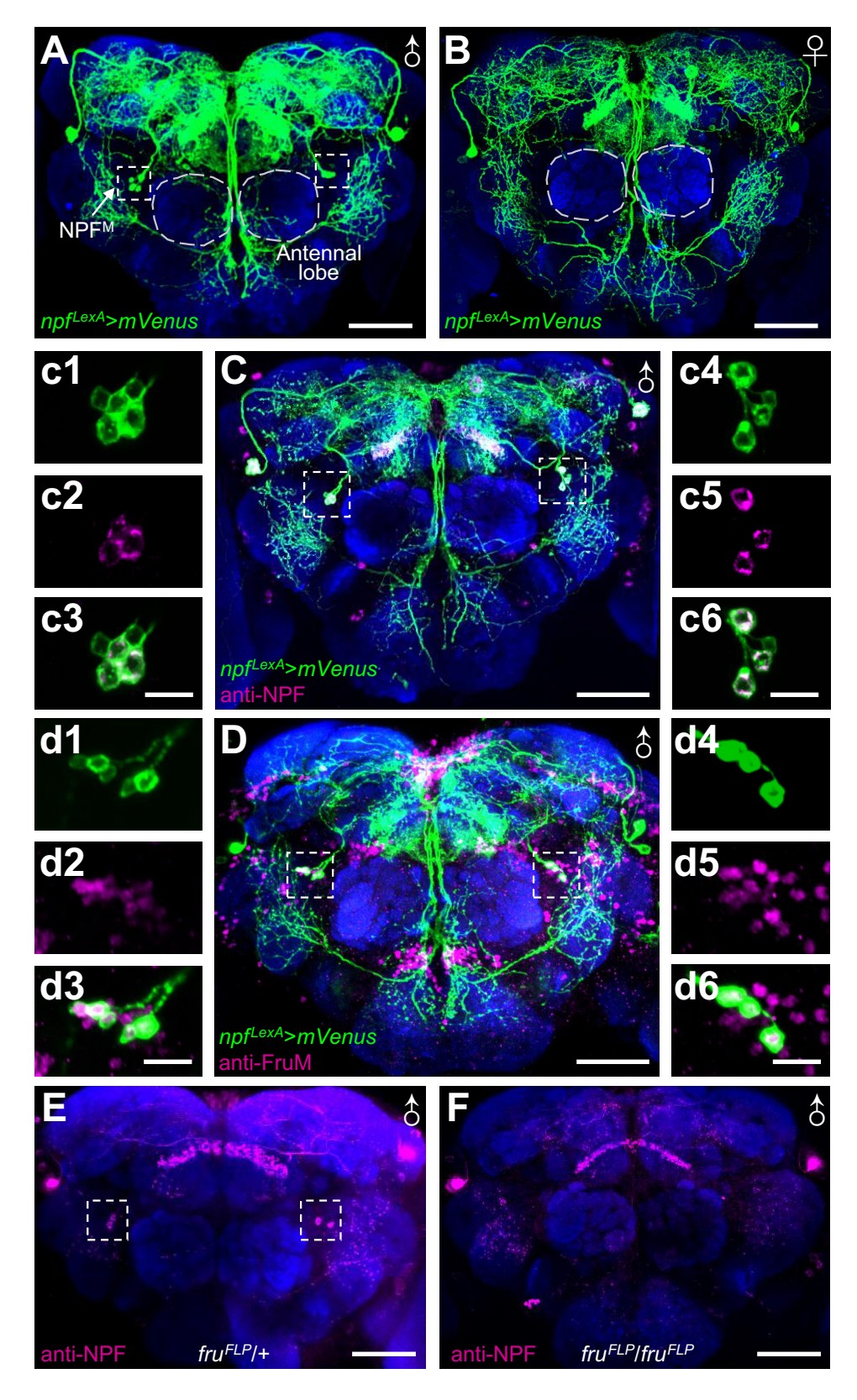

**Figure 2.** Identification of male-specific NPF[M] neurons. (**A and B**) *npf[LexA]*/LexAop-IVS-mVenus male and female brains immunostained with anti-GFP to detect mVenus. The boxes indicate NPF[M] neurons, and the circles indicate the antennal lobes. (**C**) *npf[LexA]*/LexAop-IVS-mVenus male brain immunostained with anti-GFP and anti-NPF. NPF[M] neurons are boxed. (**c1—c6**) Zoomed in images showing NPF[M] neurons. (**D**) *npf[LexA]*/LexAop-IVS-mVenus male brain immunostained with anti-GFP and anti-FruM. The boxes indicate NPF[M] neurons. (**d1—d6**) Zoomed in images showing NPF[M] neurons. (**E and F**) *fru* mutant (*fru[FLP]*/*fru[FLP]*) and control *fru[FLP]*/+ male brains immunostained with anti-NPF. The boxes indicate NPF[M] neurons. The scale bars in A—F represent 50 μm. The scale bars in c1—c6 and d1—d6 represent 10 μm.

DOI: https://doi.org/10.7554/eLife.49574.018

The following figure supplement is available for figure 2:

**Figure supplement 1.** *w[1118-CS]* male flies stained with anti-NPF and anti-DsxM.

DOI: https://doi.org/10.7554/eLife.49574.019

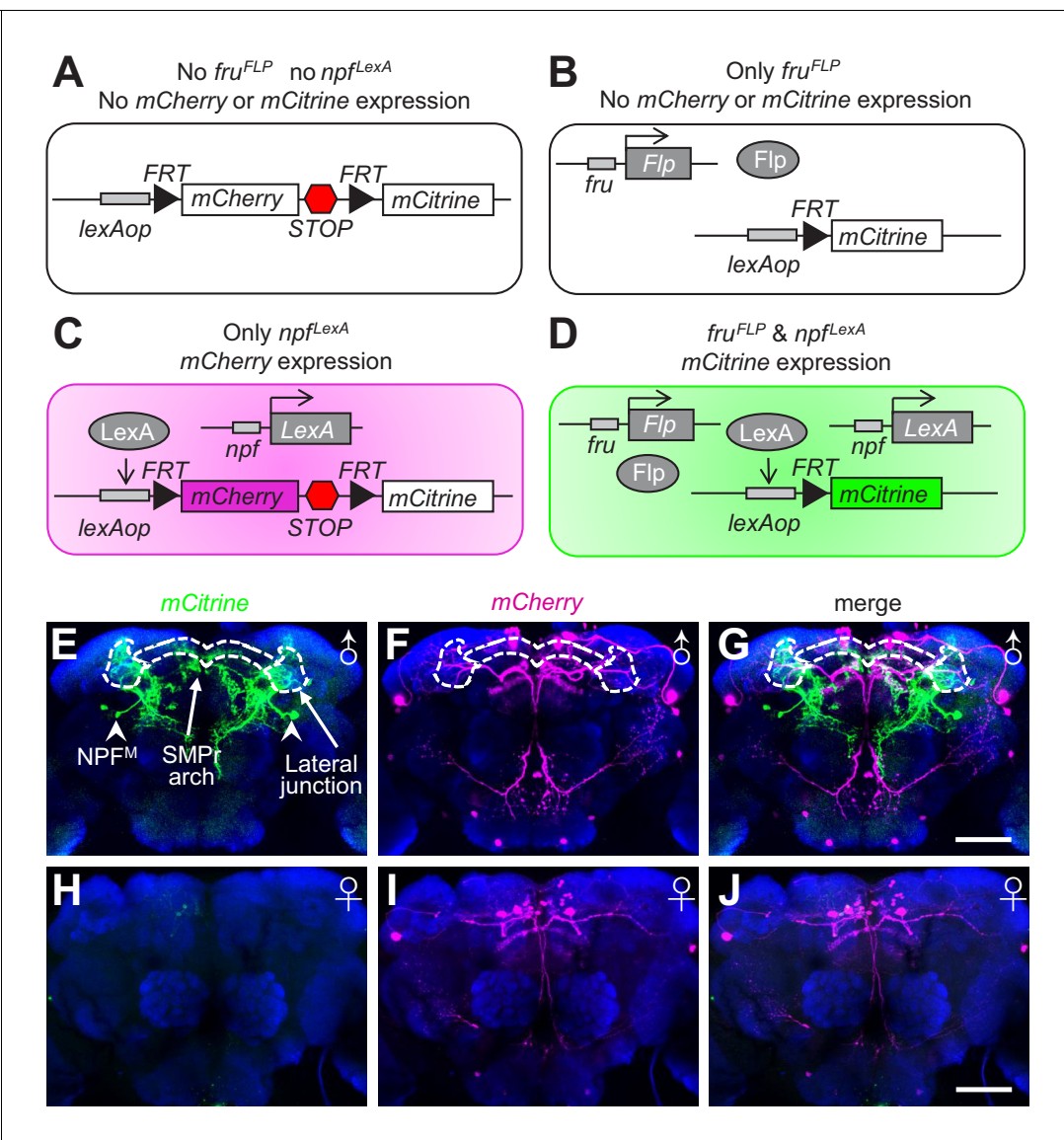

**Figure 3.** Labeling of NPF[M] neurons using the FlpOut method. (**A—D**) Schematic illustration of the FlpOut method to label NPF[M] neurons. Only neurons that express both *fru* (*fru[FLP]*) and *npf* (*npf[LexA]*) will express *mCitrine*. (**E—G**) Expression patterns of *mCitrine* (stained with anti-GFP) and *mCherry* (stained with anti-DsRed) in a male brain. The white dashes outline the SMPr arch and lateral junction regions of the LPC. Arrowheads indicate NPF[M] soma. (**H—J**) *mCitrine* and *mCherry* expression patterns in a female brain. The scale bars represent 50 μm.

DOI: https://doi.org/10.7554/eLife.49574.020

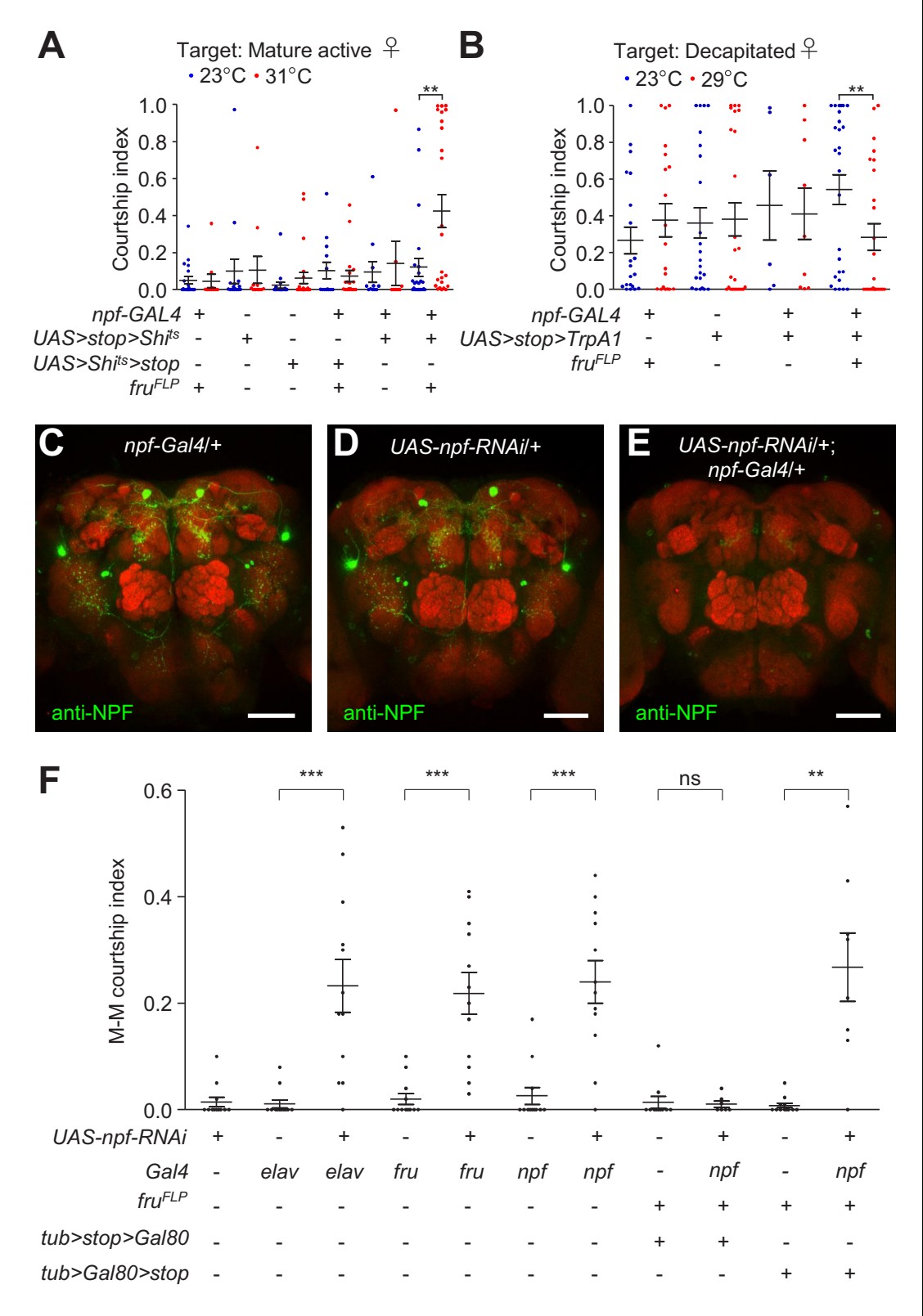

**Figure 4.** Specificity of NPF[M] neurons in regulating male courtship. (**A**) Single tester males of the indicated genotypes were assayed for male-female (M–F) courtship at both permissive (23°C) and non-permissive (31°C) temperatures for Shi[ts]. Newly-eclosed male flies were isolated for 5 days, after which they were housed with 5—7 $w^{1118}$ virgin female flies for 4 hr prior to the experiment. 7—15 day-old mature active mated $w^{1118}$ female flies were used as targets. The courtship index is the mean ratio of time spent by the tester male in courtship within 30 min following a 10 min incubation period. *Figure 4 continued on next page*

*Figure 4 continued*

n = 8—24. Bars indicate means ± SEMs. Significance was determined using Mann-Whitney test. **p < 0.01. (B) Single tester males of the indicated genotypes were assayed for courtship at two different temperatures (23℃ and 29℃). Newly-eclosed males that were isolated for 2 days were used as testers. Decapitated $w^{1118}$ female flies were used as the targets. Courtship index represents the mean ratio of time the male flies spent in courting within 10 min following a 5 min incubation period. n = 6—27. Bars indicate means ± SEMs. Significance was determined using Mann-Whitney test. **p < 0.01. (C—E) Immunohistochemistry showing the effect of *npf RNAi* knock down on NPF protein expression in male brains. Control genotypes of *npf-Gal4/+* and *UAS-npf-RNAi* male brains and experimental genotype of *npf-Gal4/+;UAS-npf-RNAi/+* male brains were immuno-stained with anti-NPF. Scale bars indicate 50 μm. (F) Effects on male-male (M–M) courtship due to *RNAi* knock down of *npf* in all neurons (*elav*), *fru* neurons, *npf* neurons, non-NPF^M *npf* neurons and NPF^M neurons. n = 7—12. The bars indicate means ± SEMs. Mann-Whitney test was used to determine significance. **p < 0.01.

DOI: https://doi.org/10.7554/eLife.49574.022

The following source data is available for figure 4:

**Source data 1.** *Figure 4A* Source data.
DOI: https://doi.org/10.7554/eLife.49574.023
**Source data 2.** *Figure 4A* Summary statistics.
DOI: https://doi.org/10.7554/eLife.49574.024
**Source data 3.** *Figure 4B* Source data.
DOI: https://doi.org/10.7554/eLife.49574.025
**Source data 4.** *Figure 4B* Summary statistics.
DOI: https://doi.org/10.7554/eLife.49574.026
**Source data 5.** *Figure 4F* Source data.
DOI: https://doi.org/10.7554/eLife.49574.027
**Source data 6.** *Figure 4F* Summary statistics.
DOI: https://doi.org/10.7554/eLife.49574.028

transgenes, then *mCherry* is removed by FlpOut in *fru*-expressing neurons, and *mCitrine* is expressed (*Figure 3D*). Therefore, NPF^M are the only neurons labeled with mCitrine. Using this intersectional method, we found that NPF^M neurons extensively arborize a large proportion of the superior brain of the male (*Figure 3E—G* and *Video 1*). In contrast, there were no mCitrine-labeled neurons in the female brain (*Figure 3H*). Rather, the NPF neurons in females were labeled with *mCherry* only (*Figure 3I and J*).

To address whether NPF^M neurons are exclusively responsible for regulating male courtship, we expressed a conditional repressor or activator specifically in NPF^M neurons. To inhibit NPF^M neurons, we used the temperature sensitive Shi^ts, which we expressed in NPF^M neurons only using the FlpOut method. We employed a transgene that encodes *Shi^ts* downstream of a 5' transcriptional *stop* cassette that is flanked by FRT sites ( *UAS > stop > Shi^ts* ; note that ' >" indicates FRT sites), and removed the *stop* cassette specifically in *fru* neurons by expressing flippase exclusively in *fru* neurons with the *fru^FLP*. After the *stops* are removed, we expressed *UAS-Shi^ts* under control of the *npf-Gal4*, thereby restricting Shi^ts to NPF^M neurons only. When we performed courtship assays at the non-permissive temperature for Shi^ts (31 ℃), the males showed elevated courtship relative to flies with the same genotype that were assayed at the permissive temperature (23℃) for Shi^ts (*Figure 4A*). We then tested the effects of inhibiting neurons except for NPF^M neurons, using *npf-Gal4/+;fru^FLP/ UAS > Shi^ts >* stop flies, which removes Shi^ts just in NPF^M neurons. These males displayed similar levels of courtship at both the permissive temperature and non-permissive temperatures for Shi^ts (*Figure 4A*).

To activate NPF^M neurons, we employed a similar FlpOut approach, using *UAS > stop > trpA1/npf-Gal4; fru^FLP/ +*flies, to express the thermally-activated TRPA1-A isoform in NPF^M neurons only. This TRPA1 isoform is a Na^+ and Ca^{2+}-permeable channel, which is activated at temperatures above ~ 27 ℃ (*Viswanath et al., 2003*). To perform these assays, we used decapitated females since intact females stimulate ceiling levels of male courtship, which are resistant to down-regulation, while decapitated females induce moderate levels, which facilitate detecting subtle decreases in male courtship. We found that courtship levels in *UAS > stop > trpA1/npf-Gal4;fru^FLP/+* males were suppressed at 29℃ relative to 23℃ (*Figure 4B*). In contrast, none of the three types of control flies exhibited lower male courtship at 29℃ (*Figure 4B*). Nevertheless, because the CI exhibited by the *UAS > stop > trpA1/npf-Gal4;fru^FLP/+* males at 29℃ was not elevated relative to the CIs displayed

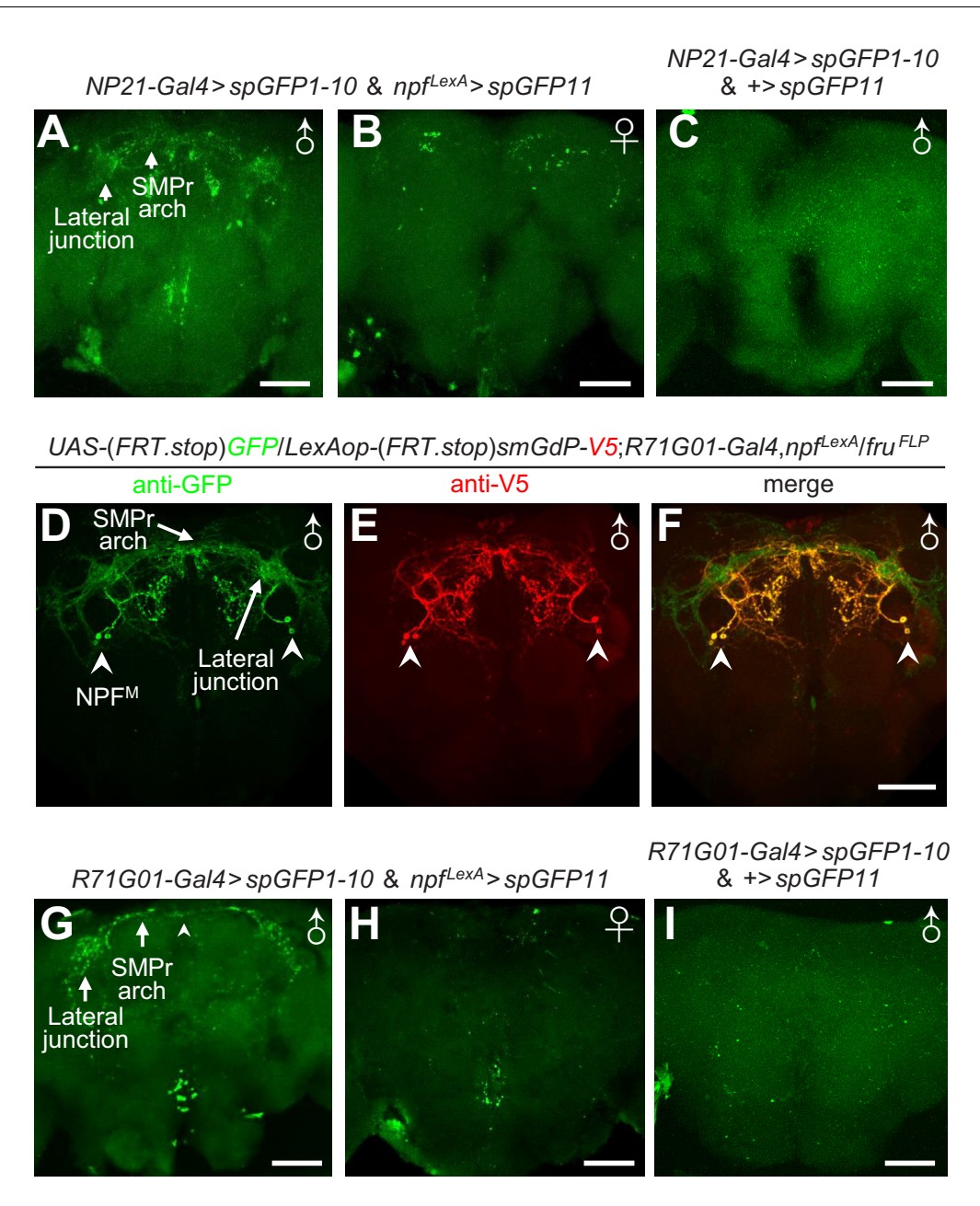

**Figure 5.** Anatomical and functional interactions between P1 and NPF[M] neurons. (**A—C**) GRASP approach to examine close interactions between NPF and *fru* neurons in *UAS-spGFP1-10, LexAop-spGFP11/NP21-Gal4, npf[LexA]* flies. GFP fluorescent signals indicate close associations. (**A**) Reconstituted GFP signals in a male brain. The arrows indicate the SMPr arch and lateral junction structures. (**B**) Reconstituted GFP signals in a female brain. (**C**) Negative control for GRASP showing a *UAS-spGFP1-10, LexAop-spGFP11/NP21-Gal4* male brain. Scale bars indicate 50 μm. A portion of the brain stacks, including the LPC structure, is shown. The full brain stacks are presented in the source data files. (**D—F**) FlpOut approach to differentially label P1 neurons and NPF[M] neurons. (**D**) Anti-GFP stained *fru*-positive P1 (due to *smGdP* expression) and NPF[M] neurons. Arrows indicate the SMPr arch and the lateral junction. Arrowheads indicate the soma of NPF[M] neurons. (**E**) Anti-V5 exclusively labels NPF[M] neurons. The arrowheads indicate soma of NPF[M] neurons. (**F**) Composite of P1 and NPF[M] neurons. The arrowheads indicate NPF[M] soma. The scale bar represents 50 μm. (**G—I**) GRASP approach to examine close interactions between NPF and P1 neurons in *UAS-spGFP1-10,LexAop-spGFP11/R71G01-Gal4,npf[LexA]* flies. GFP fluorescent signals indicate close associations. (**G**) Reconstituted GFP signals in a male brain. Arrows indicate lateral junction and SMPr arch of the LPC. The arrowhead indicates an example of a reconstituted GFP signal. (**H**) Reconstituted GFP signals in a female brain. (**I**) Negative control for GRASP showing a *UAS-spGFP1-10, LexAop-spGFP11/R71G01-Gal4* male brain. Scale bars indicate 50 μm. A portion of the brain stacks, including the LPC structure, is shown. The full brain stacks are presented in the source data files.

DOI: https://doi.org/10.7554/eLife.49574.029

*Figure 5 continued on next page*

*Figure 5 continued*

The following source data and figure supplements are available for figure 5:

**Source data 1.** *Figure 5A—C, G—I* Full stacks.
DOI: https://doi.org/10.7554/eLife.49574.033
**Figure supplement 1.** Cartoons of male brains showing the approximate positions of selected brain regions and neurons.
DOI: https://doi.org/10.7554/eLife.49574.030
**Figure supplement 2.** Comparison of the projection patterns of NPF and P1 neurons in a male brain.
DOI: https://doi.org/10.7554/eLife.49574.031
**Figure supplement 3.** Directionality of connections between P1 and NPF$^M$ neurons.
DOI: https://doi.org/10.7554/eLife.49574.032

by the control males at 29°C, the results preclude the conclusion that activation of sexually dimorphic NPF$^M$ neurons inhibits male courtship.

To test whether the NPF produced in NPF$^M$ neurons is responsible for inhibiting male courtship, we knocked down NPF expression in distinct groups of neurons. To conduct these experiments, we used *UAS-npf-RNAi*, which was effective as it greatly reduced NPF levels (*Figure 4C—E*). We found that knocking down *npf* expression with the *fru-Gal4* induced a dramatic increase in M—M courtship, and did so to a similar extent as when we used a pan-neuronal (*elav*) *Gal4* or the *npf-Gal4* (*Figure 4F*).

To specifically interrogate a requirement for NPF in NPF$^M$ neurons, we used FlpOut to introduce *Gal80* (which binds and inhibits *Gal4* activity) in either *fru$^+$* or *fru$^-$* neurons, thereby confining *UAS-npf-RNAi* expression to *fru$^-$* NPF neurons or NPF$^M$ neurons, respectively. To knockdown *npf* specifically in NPF$^M$ neurons, we used the following flies that cause excision of *Gal80* in *fru* neurons only, thereby allowing *Gal4* expression and RNA knockdown in NPF$^M$ neurons: *npf-Gal4/tub > Gal 80 >stop;UAS-npf-RNAi/fru$^{FLP}$* flies. Conversely, to prevent *npf* knockdown in NPF$^M$ neurons, we expressed *Gal80* specifically in these neurons using *npf-Gal4/tub > stop > Gal80;UAS-npf-RNAi/ fru$^{FLP}$* flies. We found that knocking down *npf* exclusively in NPF$^M$ neurons elevated M—M courtship while *npf* knock down in *fru$^-$* NPF neurons did not change the level of male courtship (*Figure 4F*). These results indicate that sexually dimorphic NPF$^M$ neurons are the subset of NPF neurons that are exclusively required for suppressing male courtship, and the effect is dependent on NPF produced in NPF$^M$ neurons.

## P1 neurons directly activate NPF$^M$ neurons

To address how NPF$^M$ neurons are integrated into the *fru* circuit, we adopted the GRASP (GFP Reconstitution Across Synaptic Partners) (*Feinberg et al., 2008*; *Gordon and Scott, 2009*) method to detect potential contact loci between NPF neurons and *fru* neurons. This approach employs a dual binary expression system to synthesize two complementary but non-functional parts of GFP (spGFP1-10 and spGFP11) on the cell membranes of distinct neurons. When the neurons are in close proximity, GFP is reconstituted and fluorescence is produced. We expressed spGFP1-10 and spGFP11 in *fru* and NPF neurons, respectively and detected strong bouton-shaped GFP signals in the male brain (*Figure 5A*) but only sparse signals in the female brain (*Figure 5B*) and no specific reconstituted GFP signals in control male brains missing the driver for the *LexAop-spGFP11* (*Figure 5C*). The reconstituted GFP signals in the male brain reconstruct a distinctive male-specific brain structure – the lateral protocerebral complex (LPC), which includes several neuropils: the lateral junction, superior medial protocerebrum (SMPr) arch, lateral crescent and the ring structure (*Figure 5—figure supplement 1A*) (*Yu et al., 2010*). The LPC structure is formed by neural projections from a cluster of male-specific P1 neurons which function as the integrative hub controlling male courtship behavior (*Kimura et al., 2008*; *Yu et al., 2010*; *Kohatsu et al., 2011*; *von Philipsborn et al., 2011*; *Pan et al., 2012*; *Bath et al., 2014*; *Inagaki et al., 2014*; *Clowney et al., 2015*; *Kallman et al., 2015*; *Zhou et al., 2015*; *Zhang et al., 2016*).

To compare the projection patterns of NPF and P1 neurons, we expressed GFP and RFP in NPF and P1 neurons, respectively, using two binary expression systems. We found that the projections from NPF neurons overlapped with the LPC structure in the male brain (*Figure 5—figure supplement 2A—C*). However, the female brain does not include an LPC structure (*Figure 5—figure supplement 2D—F*). We further combined the FlpOut method and dual binary expression systems to

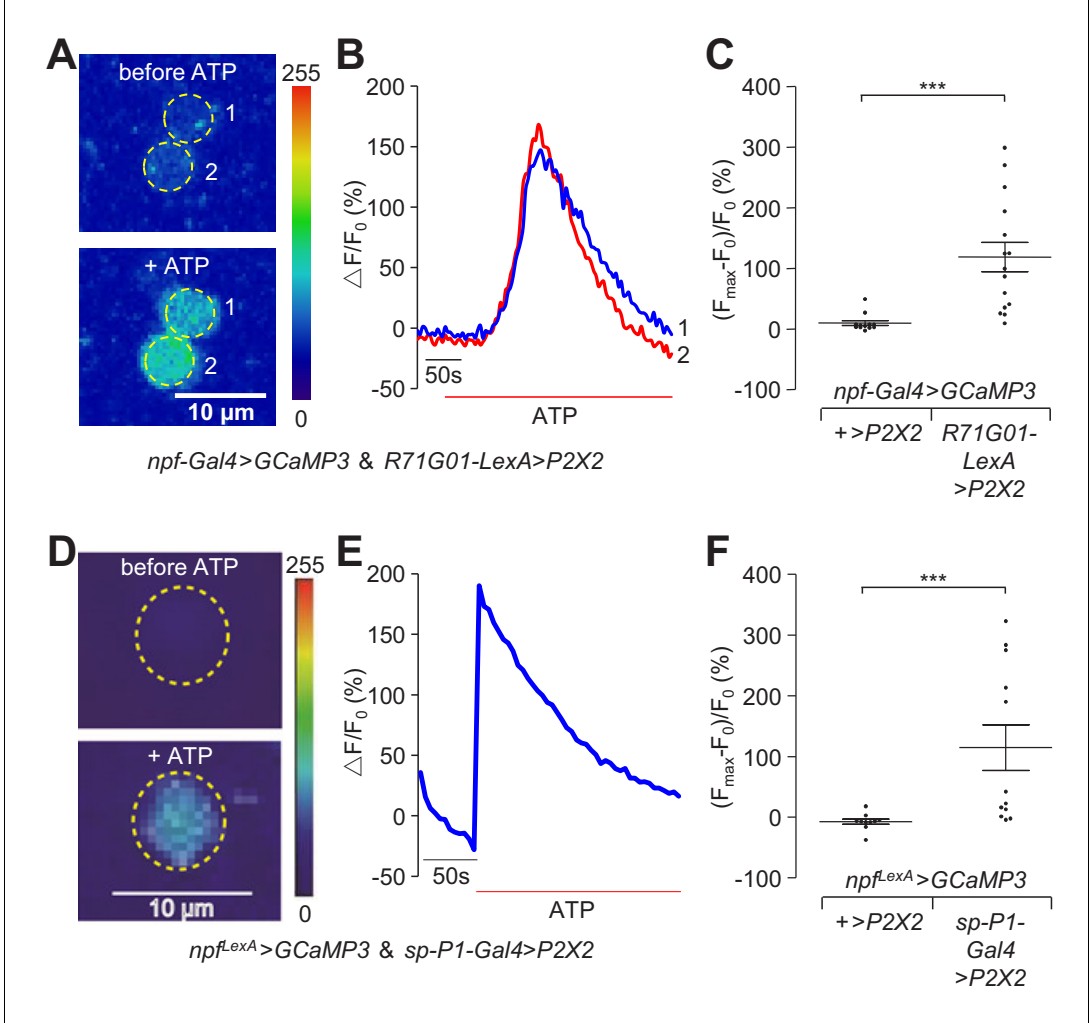

**Figure 6.** Neural activity changes in NPF[M] neurons in response to activation of P1 neurons. (**A—C**) *UAS-GCaMP3, LexAop- P2X2/R71G01-LexA;npf-Gal4/+* male brains were imaged for GCaMP3 responses. Cell bodies of NPF[M] neurons were imaged. (**A**) Representative heat maps indicating GCaMP3 fluorescence before and during ATP application. The numbers indicate NPF[M] neurons. (**B**) Representative traces showing dynamic changes in GCaMP3 fluorescence in NPF[M] neurons (circled in panel **A**). (**C**) Largest GCaMP3 fluorescence changes [($F_{max}$-$F_0$)/ $F_0$ (%)] in response to ATP application in the control and experimental group. GCaMP3 fluorescence was recorded from 12 NPF[M] neurons from eight control brains, and 15 NPF[M] neurons from nine experimental brains. (**D—F**) *UAS- P2X2 , LexAop-GCaMP3/R15A01-AD; npf[LexA] / R 71 G01-DBD* male brains were imaged for GCaMP3 responses. The cell bodies of NPF[M] neurons were imaged. (**D**) Representative heat maps indicating GCaMP3 fluorescence before and during ATP application. The numbers indicate NPF[M] neurons. (**E**) Representative traces showing dynamic changes in GCaMP3 fluorescence in NPF[M] neurons (circled in panel **D**). (**F**) Largest GCaMP3 fluorescence changes [($F_{max}$-$F_0$)/ $F_0$ (%)] in response to ATP application in the control and experimental group. GCaMP3 fluorescence was recorded from 10 NPF[M] neurons from three control brains, and 12 NPF[M] neurons from three experimental brains. The scale bars in (**A** and **D**) represent 10 μm. The bars in (**C** and **F**) indicate means ± SEMs. Significance was assessed using the Mann Whitney test, ***p < 0.001.

DOI: https://doi.org/10.7554/eLife.49574.035

The following source data and figure supplement are available for figure 6:

**Source data 1.** *Figure 6C* Source data.
DOI: https://doi.org/10.7554/eLife.49574.037
**Source data 2.** *Figure 6C* Summary statistics.
DOI: https://doi.org/10.7554/eLife.49574.038
**Source data 3.** *Figure 6F* Source data.
DOI: https://doi.org/10.7554/eLife.49574.039
**Source data 4.** *Figure 6F* Summary statistics.
DOI: https://doi.org/10.7554/eLife.49574.040
**Figure supplement 1.** Ca[2+] imaging of NPF[M] neurons in response to activation of P1 neurons.
DOI: https://doi.org/10.7554/eLife.49574.036

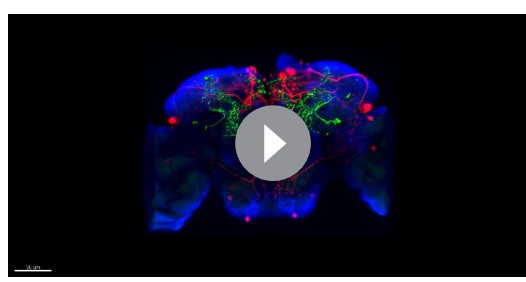

**Video 1.** Morphology of male-specific NPF[M] neurons. A male brain from a *LexAop > mCherry > mCitrine/+; npf[LexA]/fru[FLP]* fly was stained with anti-GFP (recognizes mCitrine) and anti-DsRed (recognizes mCherry). The *npf* and *fru* double positive (NPF[M]) neurons express mCitrine and are labeled by anti-GFP (*Figure 3E—G*). The remaining *npf* neurons are labeled by mCherry, and are stained with anti-DsRed. Imaris (Bitplane) software was used to prepare the reconstruction and animation.
DOI: https://doi.org/10.7554/eLife.49574.021

exclusively label NPF[M] and P1 neurons, and found that the projections from these two clusters of neurons overlapped intensely in LPC region (*Figure 5D—F* and *Figure 5—figure supplement 1* and *Video 2*).

To address if the projections of NPF and P1 neurons form direct connections, we used the *R71G01-Gal4* (which is expressed in P1 neurons and a few other neurons) to drive expression of spGFP1-10, and *npf[LexA]* to drive expression of spGFP11. We detected strong GFP signals reconstructing the LPC structure in the male brain (*Figure 5G*), but not in the corresponding brain regions of female brains or control male brains that do not have the driver for *LexAop-spGFP11* (*Figure 5H and I*). The GRASP GFP signals appear to be due to expression of the two parts of the split GFP in NPF[M] and P1 neurons for the following reasons. First, NPF[M] and P1 neurons are both male-specific, and the GRASP signals are primarily in the male brain and not in the female brain (*Figure 5G and H*). Second, the GRASP signals label two LPC structures: the lateral junction and SMPr arch (*Figure 5G*). Third, the projections of NPF[M] and P1 overlap extensively in the lateral junction and SMPr arch (*Figure 5D—F* and *Video 2*), while *fru[-]* NPF projections do not innervate the LPC region (*Figure 3F and G* and *Video 1*). Thus, the GRASP signals in the LPC structure appear to be formed by connections between NPF[M] and P1 neurons.

To clarify the directionality of the synaptic connections between NPF[M] and P1 neurons, we employed genetically encoded markers to label the dendritic (*UAS-DenMark*) and axonal (*UAS-syt:: eGFP*) branches of NPF and P1 neurons (*Wang et al., 2007*; *Nicolaï et al., 2010*). The P1 neurons that extend processes to the lateral junction and SMPr arch within the LPC structure were stained with both Denmark and Syt::eGFP, suggesting that P1 neurons send and receive signals within these neuropils (*Figure 5—figure supplement 3A, a1-a6*). However, in the corresponding lateral junction and SMPr arch within the LPC region, NPF neurons were labeled with DenMark only (*Figure 5—figure supplement 3B, b1-b3*), suggesting that NPF neurons mainly receive signals within this region. The NPF axons that stained with Syt::eGFP occurred in several brain regions other than the LPC region (*Figure 5—figure supplement 3B, b1-b6*).

To distinguish the boutons formed by NPF[M] neurons from other NPF neurons, we used the FlpOut approach to specifically label projections of NPF[M] neurons. We stained the brains of male *UAS > stop > mCD8::GFP/+;fru[FLP]/npf-Gal4* flies (*Yu et al., 2010*) with anti-GFP and anti-NPF so that the boutons formed by NPF[M] neurons would be double labeled. We found that the double-labeled boutons were concentrated in the medial anterior brain, but not in the lateral superior brain (*Figure 5—figure supplement 3C, c1-c6*), indicating that the release site of NPF[M] neurons was outside the LPC region. These results demonstrate that NPF[M] neurons do not directly act on P1 neurons. Rather, the synaptic connections between NPF[M] and P1 neurons in the LPC region are formed by pre-

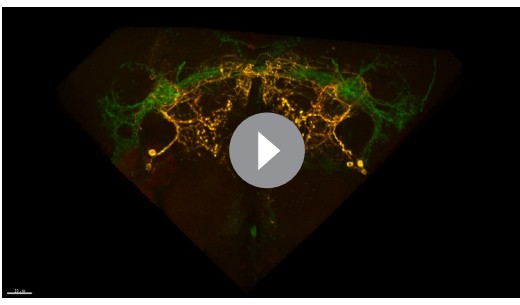

**Video 2.** Animated representation of projections of NPF[M] and P1 neurons. *UAS > stop > mCD8::GFP/ LexAop > stop > myr::smGdP-V5;R71G01-Gal4,npf[LexA]/ fru[FLP]* male brain stained with anti-GFP and anti-V5. The P1 neurons were singly labeled with anti-GFP, while NPF[M] neurons were double labeled with anti-GFP and anti-V5. Imaris (Bitplane) software was used to prepare the reconstruction and animation.
DOI: https://doi.org/10.7554/eLife.49574.034

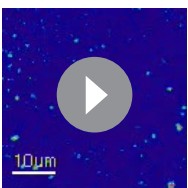

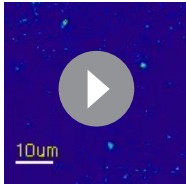

**Video 3.** Activation of P1 neurons causes a significant increase in GCaMP3 fluorescence in NPF[M] neurons. The NPF[M] neurons were imaged in a *UAS-GCaMP3, LexAop P2X2/R71G01-LexA;npf-Gal4/+* male brain. ATP was applied to the brain sample as indicated. DOI: https://doi.org/10.7554/eLife.49574.041

**Video 4.** Activation of P1 neurons causes a significant increase in GCaMP3 fluorescence in NPF[M] neurons following application of ATP to the brain sample. The NPF[M] neurons were imaged in a *UAS- P2X2, LexAopGCaMP3/+;R71G01-Gal4,npf[LexA]/+* male brain. ATP was applied to the brain sample as indicated. DOI: https://doi.org/10.7554/eLife.49574.042

synaptic P1 neurons and post-synaptic NPF[M] neurons.

To determine the impact of activation of P1 neurons on the activity of the NPF[M] neurons, we combined chemogenetics and GCaMP imaging to monitor $Ca^{2+}$ dynamics (*Yao et al., 2012*) as an indicator of neural activation. We expressed *P2X2* (encoding an ATP-gated cation channel) (*Lima and Miesenböck, 2005*) in P1 neurons, and expressed *GCaMP3* in NPF neurons. We used *R71G01-LexA*, which is expressed in P1 neurons and a few other neurons, to drive *P2X2* expression, and *npf-Gal4* to drive *UAS-GCaMP3*. In a complementary experiment, we switched the two binary systems, and used the *R71G01-Gal4* and *npf[LexA]* to drive *P2X2* and *GCaMP3*, respectively. Because the diffusion rate and final concentration of ATP that reaches the brain varies across samples, we calculated the maximum fold changes of the GCaMP3 responses after ATP application relative to the basal levels of GCaMP3 before ATP application. We found that ATP-induced activation of P1 neurons led to robust GCaMP3 signals in NPF[M] neurons (*Figure 6A—C* and *Figure 6—figure supplement 1* and *Videos 3* and *4*).

In order to exclude the impact from other neurons, we expressed *P2X2* in P1 neurons only using a *split-P1-Gal4* comprised of *R15A01-AD* (activation domain) and *R71G01-DBD* (DNA-binding domain). We imaged $Ca^{2+}$ dynamics in NPF[M] neurons in response to ATP application, and detected large increases in GCaMP3 fluorescence in response to activation of P1 neurons (*Figure 6D—F*), further supporting the conclusion that P1 neurons directly activate NPF[M] neurons.

## Increase in courtship by inhibiting NPF[M] neurons depends indirectly on P1 neurons

To determine whether the function of NPF[M] neurons in courtship regulation is dependent on P1 neurons, we tested if silencing P1 neurons would prevent the courtship elevation induced by disruption of NPF neurons. We expressed *UAS-Shi[ts]* in both NPF and P1 neurons (*npf-Gal4* and *R71G01-Gal4*) and assayed male courtship at both permissive and non-permissive temperatures. We found that the courtship dis-inhibition caused by disrupting NPF neurons was eliminated by simultaneous disruption of P1 neurons (*Figure 7A—C*). The results suggest that NPF[M] neurons appear to act through P1 neurons to regulate male courtship. Alternatively, NPF[M] and P1 neurons may act in parallel and serve opposing inputs onto a common neuronal target.

NPF binds to a G protein-coupled receptor—the NPF receptor (NPFR), which couples to a Gi signaling pathway to inhibit *npfr*-expressing neurons (*Garczynski et al., 2002*). To address the roles of the *npfr* gene and NPFR neurons in regulating male courtship, we replaced a portion of the *npfr* coding region with *LexA*, thereby generating an *npfr* mutant and a reporter (*Figure 7—figure supplement 1*). We then used the *R71G01-Gal4* and *npf[LexA]/+* to label P1 neurons and NPFR neurons with GFP and mCherry, respectively. We found that they primarily stain distinct neuronal populations (*Figure 7D—F*), indicating that P1 neurons are not the *npfr*-expressing neurons. These results further support our data suggesting that NPF[M] axons do not send signals directly to P1 dendrites, but that P1 neurons signal to NPF[M] neurons.

We assayed courtship behavior of *npfr[LexA]* mutant flies, demonstrating that these mutant animals raised in isolation exhibit significantly higher M–M courtship than control males (*Figure 7G*). We observed similar results with *npfr[LexA]/npfr[c01896]* trans-heterozygous flies (*Figure 7G*). RNAi-mediated

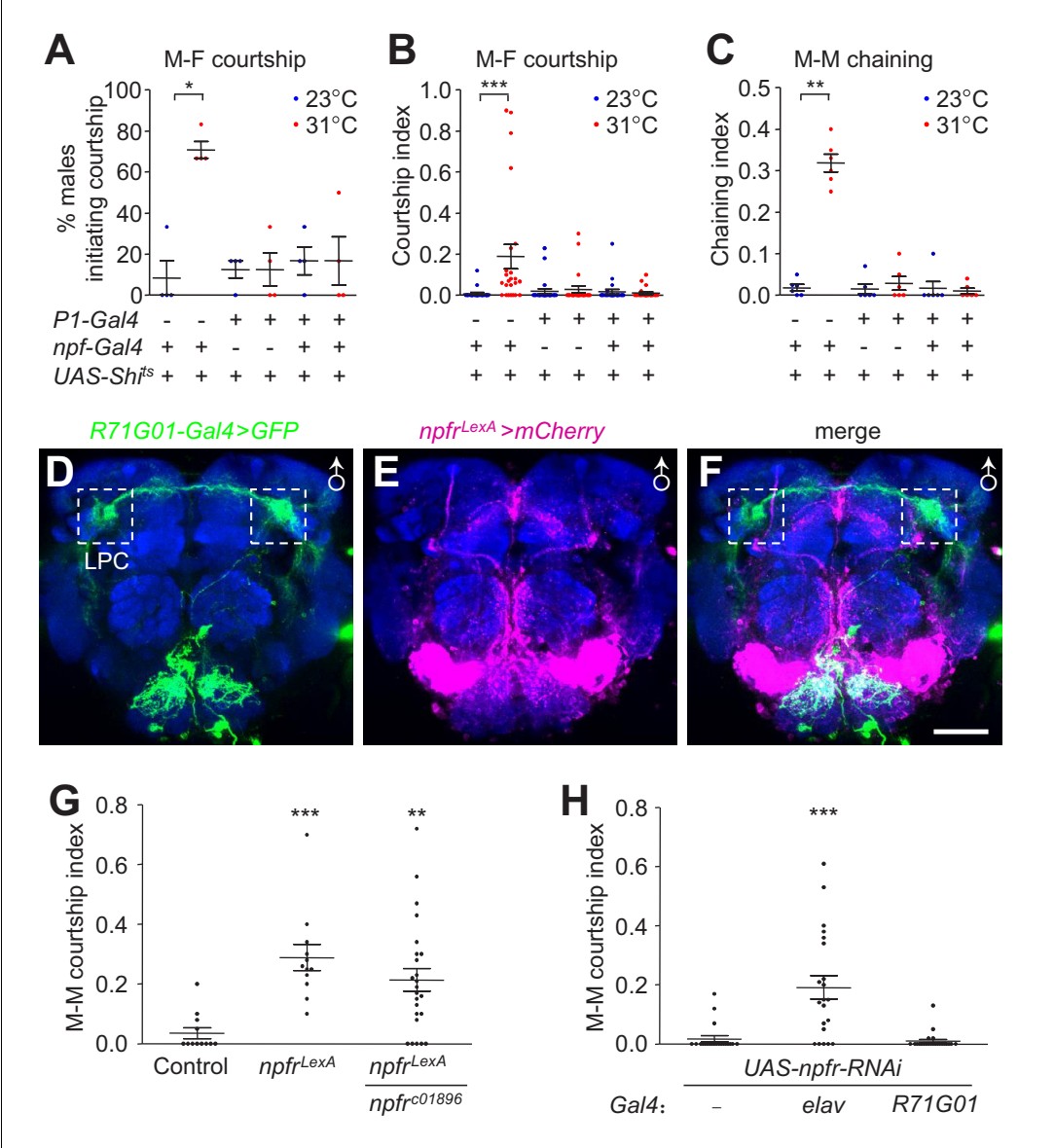

**Figure 7.** Effects of inactivating NPF and P1 neurons on male courtship, characterization of *npfr* reporter expression, and impact of *npfr* on male courtship. (A—C) Effects of silencing both NPF and P1 neurons with Shi[ts] (*npf-Gal4/+;R71G01-Gal4/UAS-Shi[ts]*) on courtship of group-housed males towards female targets. Male-female (M–F) courtship was assayed at the permissive (23°C) and non-permissive (31°C) temperatures for Shi[ts]. (A) The percentages of males that initiated courtship. n = 4 (6 flies/group). (B) The courtship indexes were scored based on 20—30 min of observation during a 30 min incubation period. n = 24. (C) Effect of silencing both NPF and P1 neurons with Shi[ts] (*npf-Gal4/+;R71G01-Gal4/UAS-Shi[ts]*) on male-male (M–M) courtship. Isolation-housed males were assayed for chaining behavior at 23°C and 31°C for 10 min. n = 6 (8—12 flies/group). The bars indicate means ± SEMs. Significance was assessed using the Mann-Whitney test. *p < 0.05, **p < 0.01, ***p < 0.001. (D—F) Spatial distribution of *npfr* (mCherry) and *P1* (GFP) reporters in a male brain (*UAS-mCD8::GFP/+;R71G01-Gal4/npfr[LexA],LexAop-mCherry*). The reporters were detected with GFP and DsRed antibodies. The boxed regions indicate the LPC. The scale bar represents 50 μm. (G) *npfr[LexA]* homozygous and *npfr[LexA]/npfr[c01896]* trans-heterozygous mutants were assayed for M–M courtship. The control flies are *w[1118-CS]*. n = 12—24. (H) Effects on M–M courtship due to knock down of *npfr* pan-neuronally (*elav-Gal4*) or in P1 neurons. n = 21—23. The bars indicate means ± SEMs. To determine significance, we used the Kruskal-Wallis test followed by the Dunn's *post hoc* test. **p < 0.01, ***p < 0.001.

DOI: https://doi.org/10.7554/eLife.49574.043

The following source data and figure supplement are available for figure 7:

**Source data 1.** *Figure 7A* Source data.
DOI: https://doi.org/10.7554/eLife.49574.045
**Source data 2.** *Figure 7A* Summary statistics.
DOI: https://doi.org/10.7554/eLife.49574.046

*Figure 7 continued on next page*

*Figure 7 continued*

**Source data 3.** *Figure 7G—H* Source data.
DOI: https://doi.org/10.7554/eLife.49574.047
**Source data 4.** *Figure 7G—H* Summary statistics.
DOI: https://doi.org/10.7554/eLife.49574.048
**Figure supplement 1.** $npfr^{LexA}$ mutant.
DOI: https://doi.org/10.7554/eLife.49574.044

knockdown of *npfr* using a pan-neuronal *Gal4* (*elav*) also increased M–M courtship behavior (*Figure 7H*). In contrast, knocking down *npfr* expression in P1 neurons had no effect (*Figure 7H*).

We took advantage of the GRASP method to investigate whether NPFR and P1 neurons make direct connections. We used *R71G01-Gal4* and $npfr^{LexA}$ drivers to express spGFP1-10 and spGFP11 respectively. We detected GRASP signals in the lateral crescent within the LPC region of the male brain (*Figure 8A,a1,a2*). In contrast, we did not detect GRASP GFP fluorescence in female brains or in control male brains (*Figure 8B,b1,b2* and *Figure 5I*).

To examine whether activation of NPFR neurons affects the activity of P1 neurons, we expressed *P2X2* in NPFR neurons, and *GCaMP3* in P1 neurons. We found that activation of NPFR neurons with ATP application induced robust GCaMP3 responses in the LPC structure (*Figure 8C—E* and *Video 5*). In control flies that did not express *P2X2*, application of ATP did not induce elevation of GCaMP3 fluorescence (*Figure 8E*). The preceding results indicate that at least a subset of NPFR neurons anatomically connect and functionally activate P1 neurons. Together, our results indicate that NPF^M, NPFR and P1 neurons form intricate interactions, and ensure proper courtship output in accordance with a male's internal drive state.

## Discussion

Multiple studies report the contribution of external sensory cues in inducing or suppressing male courtship behavior by signaling onto the P1 courtship decision center in the male brain (*Kimura et al., 2008*; *Yu et al., 2010*; *Kohatsu et al., 2011*; *von Philipsborn et al., 2011*; *Pan et al., 2012*; *Bath et al., 2014*; *Inagaki et al., 2014*; *Clowney et al., 2015*; *Kallman et al., 2015*; *Kohatsu and Yamamoto, 2015*; *Zhou et al., 2015*). In contrast, much less is known about how the P1 neurons are regulated by the male's prior mating experience (*Inagaki et al., 2014*) and how courtship is affected by the internal drive state. An exception is a recent study that identified a group of dopaminergic neurons that changes in activity in proportion to male mating drive, and which directly activates P1 neurons to promote male courtship (*Zhang et al., 2016*). In the current study, we characterized a cluster of male-specific NPF^M neurons which functions antagonistically to dopamine neurons by serving to suppress courtship by responding to sexual satiation. Disruption of NPF^M neurons causes dis-inhibition of courtship in satiated males. The internal drive state of males is encoded by opposing excitatory and inhibitory inputs, which enable a male to make an appropriate mating decision in accordance with its internal drive state.

### Suppression of NPF neurons or elimination of *npf* counters sexual satiation

Elimination of *npf* or knocking down *npf* expression exclusively in male-specific NPF^M neurons causes male flies to exhibit maladaptive, hypersexual activity. In contrast to control males, which are sexually satiated when exposed to an abundance of females, and consequently display very low courtship levels, we found that flies overcome the sexual satiation imposed by mating if we introduce a loss-of-function mutation in *npf* or inhibit NPF neurons. Thus, satiation of courtship is dis-inhibited by disrupting NPF signaling.

Our findings that suppressing or eliminating NPF neurons elevates male courtship is in contrast to a previous report that genetic disruption or feminization of NPF neurons reduces male courtship activity (*Lee et al., 2006*). Maintaining males in the presence or absence of females profoundly affects sexual satiation levels, and the housing conditions were not clearly defined in this previous study. Our conclusions are supported by multiple lines of evidence. First, we found that when we inhibit neurotransmission from NPF neurons, using a temperature sensitive dynamin (*Shi*^ts), the males

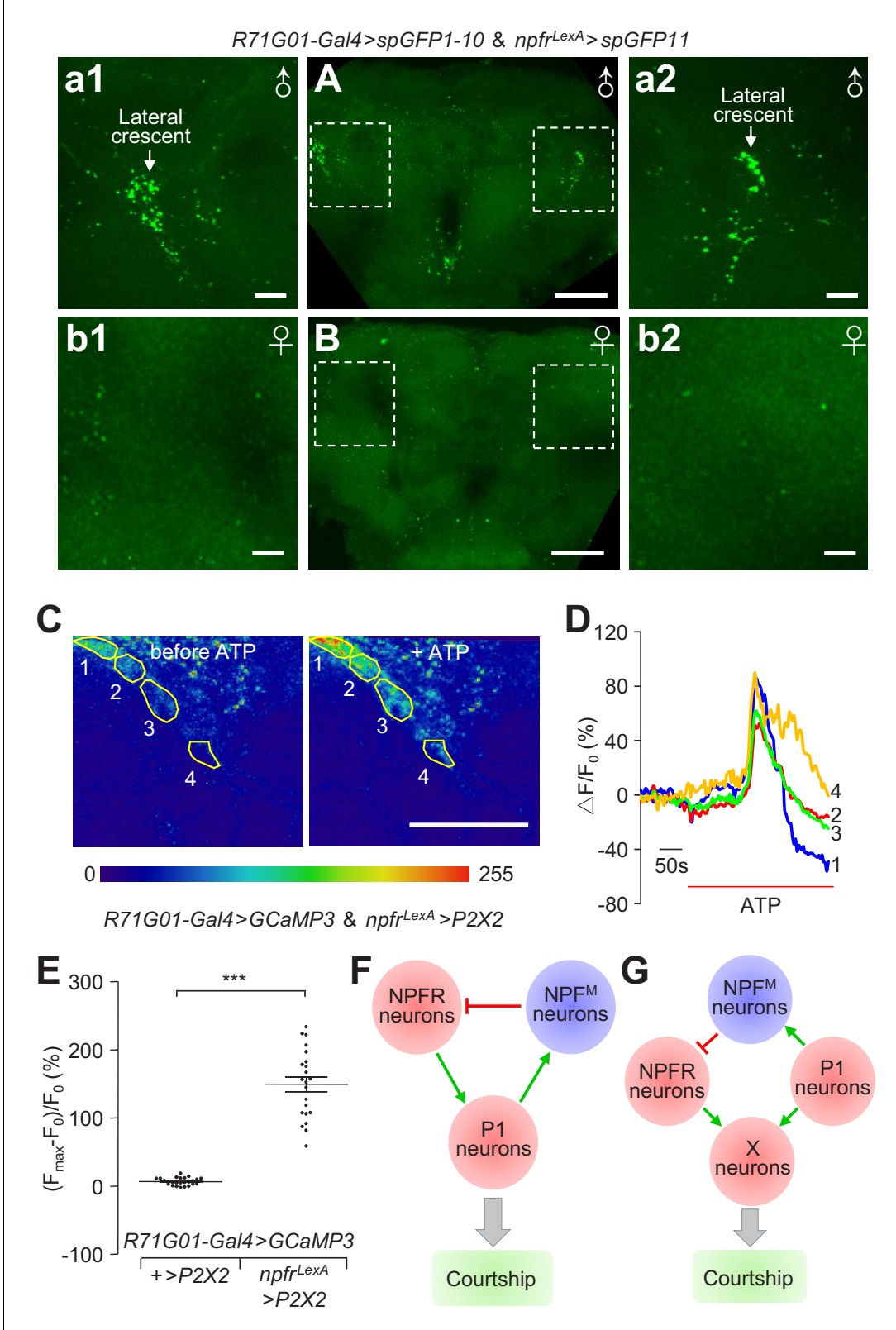

**Figure 8.** Anatomical and physiological interactions between NPFR and P1 neurons. (**A and B**) GRASP analyses to test for close associations between *npfr* and *P1* neurons. *UAS-spGFP1-10,LexAop-spGFP11/R71G01-Gal4,npfr^LexA* male and female brains were imaged for reconstituted GFP signals. (**A**) Reconstituted GFP signals in a male brain. The boxes indicate the higher magnification images (**a1 and a2**) showing the bouton-shaped GFP signals in the lateral crescent within the LPC. (**B**) Reconstituted GFP signals in a female brain. The boxes indicate the zoomed in areas (**b1 and b2**) showing the

*Figure 8 continued on next page*

*Figure 8 continued*

lateral regions of the female brain, corresponding approximately to the lateral crescent regions in the male brain. The scale bars represent 50 µm in (**A** and **B**), and 10 µm in a1—a2 and b1—b2. A portion of the brain stacks, including the LPC structure, is shown. The full brain stacks are presented in the source data files. (**C—E**) Assaying effects on P1 neuronal activity with GCaMP3, after stimulating *npfr* neurons with ATP. *GCaMP3* and *P2X2* were expressed specifically in P1 and *npfr* neurons, respectively, in the following flies: *UAS-GCaMP3, LeAop P2X2/+;R71G01-Gal4/npfr^LexA*. GCaMP3 responses were imaged in the LPC structures in male brains. (**C**) Representative heat maps indicating GCaMP3 fluorescence before and during ATP application. The numbers indicate the regions within the LPC structure measured. (**D**) Representative traces showing dynamic fluorescence changes in the specified regions circled in (**C**). (**E**) Maximal fluorescence increases [(F$_{max}$-F$_0$)/ F$_0$ (%)] in response to ATP application. GCaMP3 fluorescence was recorded from 25 regions from five control brains, and 22 regions from four experimental brains. The scale bar in (**C**) represents 50 µm. The bars in (**E**) indicate means ± SEMs. To determine significance, we used the Mann Whitney test. \*\*\*p < 0.001. (**F**) A model illustrating the feedback loop of NPF$^M$ neurons in the regulation of P1 neuronal activity. (**G**) Illustration of a feedforward parallel model, in which target neurons (X neurons) receive parallel input from P1 neurons and NPFR neurons.

DOI: https://doi.org/10.7554/eLife.49574.049

The following source data is available for figure 8:

**Source data 1.** *Figure 8E* Source data.
DOI: https://doi.org/10.7554/eLife.49574.050
**Source data 2.** *Figure 8E* Summary statistics.
DOI: https://doi.org/10.7554/eLife.49574.051
**Source data 3.** *Figure 8A—B* Full stacks.
DOI: https://doi.org/10.7554/eLife.49574.052

showed a dramatic increase in courtship towards female conspecifics. This occurred using group-housed males which normally are sexually satiated. Second, introduction of a genetically encoded toxin, or inhibition of NPF neurons by overexpression of a K$^+$ channel, also increases courtship activity. Third, when we disrupted the *npf* gene, the mutant males displayed a remarkable increase in courtship. This effect was so profound that the males courted females of another species and also displayed a great increase in M–M courtship, even though their gender preferences remained unchanged. Fourth, disruption of the *npfr* gene resulted in significant elevation in courtship, consistent with the effect of disrupting *npf*. Fifth, when we specifically silenced male-specific *fru*$^+$ NPF (NPF$^M$) neurons, male courtship behavior was elevated. In contrast, silencing *fru*$^-$ NPF neurons had no impact on male courtship. Sixth, knocking down *npf* expression exclusively in NPF$^M$ neurons increased male courtship, while knocking down *npf* in *fru*$^-$ NPF neurons had no effect.

## Neuronal circuit models entail P1 neurons activating NPF$^M$ neurons

Our anatomical, physiological and functional evidence demonstrate that P1 neurons activate NPF$^M$ neurons, and suggest potential models through which these neurons coordinate to regulate male courtship drive. According to one model, P1 and NPF$^M$ neurons form a recurrent inhibitory neuronal circuit (*Figure 8F*). Stimulation of P1 neurons activates NPF$^M$ neurons, which act through an intermediate group of NPF receptor (NPFR neurons) and feedback to inhibit P1 neurons. This recurrent inhibitory model posits that P1 neurons are strongly activated when males are exposed to many females, inducing NPF$^M$ neurons to release NPF. This neuropeptide acts on the G$_i$-coupled NPF receptor and inhibits NPFR neurons, leading to a suppression of P1 activity, and attenuation of male courtship. When the activity of P1 neurons is reduced, stimulation of NPF$^M$ neurons and NPF release are diminished. This attenuates the feedback inhibition from NPF$^M$ to P1 neurons, leading to a return of P1 neuronal activity, and male courtship drive.

We suggest that the recurrent inhibitory neuronal motif proposed here is important for maintaining proper activities of P1 neurons, thus ensuring appropriate behavioral choices that are critical for a male's reproductive success,

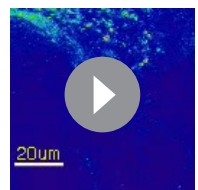

**Video 5.** Activation of NPFR neurons with ATP causes a significant increase in GCaMP3 fluorescence in the LPC structure of P1 neurons. The imaging was performed on a *UAS-GCaMP3, LexAop P2X2 /+;R71G01-Gal4/ npfr^LexA* male brain. ATP was applied to the brain sample as indicated.
DOI: https://doi.org/10.7554/eLife.49574.053

depending on the level of sexual satiety. Because P1 neurons integrate multi-modal sensory input, as well as the male's internal level of sex drive (*Kohatsu et al., 2011*; *Pan et al., 2012*; *Bath et al., 2014*; *Inagaki et al., 2014*; *Clowney et al., 2015*; *Kallman et al., 2015*; *Kohatsu and Yamamoto, 2015*; *Zhou et al., 2015*; *Zhang et al., 2016*), their activity must be under stringent control so that males display the courtship ritual only when both external sensory cues and the internal drive states are appropriate.

The recurrent inhibitory neural motif proposed here is dedicated to ensure appropriate activation of P1 neurons. Disruption of the inhibitory NPF afferents leads to excessive courtship behavior in the male fly that is maladaptive, as it overrides the courtship inhibition normally imposed by recent mating with females, other males, or females of other *Drosophila* species.

Recurrent inhibitory neural motifs are important in the central nervous system. In the mammalian spinal cord, motor neurons send collateral branches to Renshaw cells, which in turn send inhibitory signals back to motor neurons (*Alvarez and Fyffe, 2007*). The function of this recurrent inhibition is assumed to restrict excessive activation of motor neurons and contribute to precise recruitment of muscle fibers in order to generate proper force for different tasks (*Alvarez and Fyffe, 2007*). Recurrent inhibitory loops also occur in the hippocampus and entorhinal cortex. In these systems, principal cells send excitatory outputs to fast-spiking, parvalbumin-positive interneurons, and at the same time receive inhibitory inputs from these interneurons, thus, closing the feedback inhibition loop (*Pouille and Scanziani, 2004*; *de Almeida et al., 2009*; *Pastoll et al., 2013*).

While NPF[M], P1 and NPFR neurons are essential for regulating courtship by responding to prior mating experience, and may do so through a recurrent inhibitory loop (*Figure 8F*), our data do not exclude other models. Part of the argument in favor of the recurrent inhibitory loop model is that the GRASP analysis suggests that NPFR neurons make direct connections with P1 neurons. Moreover, by coupling chemogenetic manipulation and Ca$^{2+}$ imaging, we found that activation of NPFR neurons activate P1 neurons. However, NPFR neurons are widely distributed, and our data do not resolve whether the NPFR neurons that activate P1 neurons are the same subset of NPFR neurons that are the direct downstream target of NPF[M] neurons. Thus, one alternative to the recurrent inhibitory motif is a feedforward parallel model, in which target neurons (X neurons) control courtship drive by receiving parallel input from P1 neurons and NPFR neurons (*Figure 8G*). This latter model posits that P1 neurons activate X neurons, and at the same time, send axonal branches to activate NPF[M] neurons, which then act through NPFR neurons and suppress the target neurons through a feedforward mechanism. Future experiments that resolve the anatomical and functional diversity of NPFR neurons should distinguish between the recurrent inhibitory versus feedforward parallel model, which ensure proper courtship output in accordance with a male's internal drive state.

## Impact of NPF activity on courtship versus aggression

Courtship and aggression are closely interrelated social behaviors. If males are housed in isolation, they exhibit elevated courtship and aggression (*Wang et al., 2008*; *Liu et al., 2011*). This positive relationship is consistent with the observation that the presence of a potential mate promotes a male fly's propensity to fight a competitor to win a mating competition (*Kravitz and Fernandez, 2015*). Though the tendency to fight or to court is positively related, the behavioral choice between courtship and aggression is mutually exclusive.

We found that when we disrupt the activity of NPF neurons, M–M courtship is dominant over aggression. We suggest that loss of NPF function diminishes inhibition of P1 neurons. As a result, even sub-optimal stimuli strongly activate P1 neurons and induce male courtship behavior even towards inappropriate targets. Conversely, when we increase the activity of NPF neurons or overexpress the *npf*-cDNA in NPF neurons, M–M aggression is dominant over courtship.

The precise contribution of NPF neurons in regulating aggression is unresolved. One group found that activation of NPF neurons elevates male aggression (*Asahina et al., 2014*) while another reported that silencing or feminizing NPF neurons elevates aggression (*Dierick and Greenspan, 2007*). We found that when we overexpressed either the Na$^+$ channel NaChBac or the *npf-cDNA* in NPF neurons, the males exhibited increased aggression. We propose that excessive NPF activity suppresses P1 neurons, thereby setting a high threshold for P1 activation. Our observations are consistent with previous report that weaker activation of P1 neurons favors aggression while stronger activation of P1 neurons favors courtship (*Hoopfer et al., 2015*). It remains to be determined if NPF

neurons also impact on the aggression modulatory or arousal center (*Asahina et al., 2014*; *Watanabe et al., 2017*), independent of its effect on P1 neurons.

### Possible relationship of NPF to courtship regulation by mammalian NPY

NPF is the *Drosophila* counterpart of mammalian NPY, which regulates feeding, reproduction, aggression, anxiety, depression and the alcohol addiction (*Nässel and Wegener, 2011*). Previous studies indicate that sexually dimorphic NPY neurons innervate the human INAH3 (interstitial nuclei of anterior hypothalamus 3), a region correlated with sexual orientation and gender identity recognition (*LeVay, 1991*; *Byne et al., 2000*; *Garcia-Falgueras and Swaab, 2008*). The discovery that *Drosophila* NPF regulates courtship depending on the internal drive state raises questions as to whether NPY may serve similar functions in mammals.

## Materials and methods

### Key resources

Descriptions of the key fly strains, antibodies, plasmids, chemicals, kits, services and software are provided in the *Supplementary file 1*.

### Fly stocks

The following strains were obtained from Bloomington Stock Center (Indiana University): *npf-Gal4* (#25681, and #25682 have identical promoters, but are inserted on the 2$^{nd}$ and 3$^{rd}$ chromosomes, respectively), *elav-Gal4* (#8765), *fru-Gal4* (NP21 #30027), *R71G01-Gal4* (P1-Gal4 #39599), *R71G01-LexA* (P1-LexA #54733), *UAS-NaChBac* (#9468), *UAS-Kir2.1* (#6596), *UAS-DTI* (#25039), *UAS-mCD8::GFP* (#5137), *UAS-npf-RNAi* (VDRC108772), *UAS-npfr-RNAi* (VDRC107663), *UAS-DenMark,UAS-syt::eGFP* (#33064), *LexAop-mCherry* (#52271), *LexAop(FRT.mCherry)ReaChR-mCitrine* (#53744), *UAS-IVS-mCD8::RFP*, *LexAop-mCD8::GFP* (#32229), *UAS-CD4-spGFP1-10,LexAop-CD4-spGFP11* (#58755), *LexAop-IVS-CsChrimson.mVenus* (#55139), *Lexop(FRT.stop)myr::smGdP-V5* (#62107) *npfr$^{c01896}$* (#10747), *tub(FRT.Gal80)stop* (#38880), *tub(FRT.stop)Gal80* (#38878).

*UAS-npf* was a gift from Dr. Ping Shen (*Wu et al., 2003*) (University of Georgia), *UAS- P2X2,LexAop-GCaMP3* and *UAS-GCaMP3, LexAop P2X2* were from Dr. Orie Shafer (*Yao et al., 2012*) (University of Michigan), *UAS-Shibire$^{ts}$* was from Dr. Christopher Potter (*Kitamoto, 2001*) (Johns Hopkins University School of Medicine), *fru$^{FLP}$*, *UAS-(FRT.stop)mCD8::GFP*, *UAS-(FRT.stop)Shibire$^{ts}$* and *UAS-(FRT.Shibire$^{ts}$)stop*, *UAS-(FRT.stop)dTRPA1* were from Dr. Barry Dickson (*Yu et al., 2010*) (Janelia Research Campus), *R71G01-DBD;R15A01-AD* was from Dr. David Anderson (California Institute of Technology).

The *npf$^{LexA}$* and *npfr$^{LexA}$* mutants were outcrossed into a *w$^{1118}$* background for five generations. The controls for comparison to these mutants were *w$^{1118}$* flies in which we exchanged the X chromosome with Canton-S so the flies are *w$^{+}$* on the X chromosome (*w$^{1118-CS}$* flies). The full genotypes of the flies used in each figure and video are listed in *Supplementary file 2*.

### Behavioral assays

The behavioral assays were recorded using a Samsung SCB-3001 camera. All behavioral analyses were performed using these videos.

#### M–F, and M–M courtship assays

To perform courtship assays, we added 3 ml of 1.5 % agarose into each well of 24-well cell culture plates (Corning Incorporated, REF353847). 2 mm diameter holes were drilled on the cover over each well. Custom silicone plugs were prepared (435570, StockCap) for blocking the holes. The cover and the plate were taped together to avoid gaps that might allow flies to escape.

Unless otherwise specified, 5—7 days old mixed sex, group-housed males (10 males raised together with 30 virgin *w$^{1118}$* females for 3 days) were used for the courtship assays. Three types of female targets were used: 1) mature active females, 2) newly-eclosed females, or 3) decapitated females. In experiments in which the targets were either grouped-housed *w$^{1118}$* males or *Drosophila simulans* females, we used 5—7 day old isolation-housed males as the testers. One tester male and one target were ice anesthetized, and transferred together into courtship chambers. The flies were

allowed to recover for 10 min, and then male courtship was scored over the next 10 min. The courtship index is the fraction of time that a tester male performs courtship towards the target.

To test the effects of inhibiting *npf* neurons with *Shi*^ts, a single tester male (*npf-Gal4/+;UAS-Shi*^ts/+) and a target female (mature, active *w*^1118, 5—7 days old) were ice-anesthetized, and the pair was transferred into courtship chambers. The assays were performed at 23°C and 31°C, which are the permissive and non-permissive temperatures for *Shi*^ts, respectively. Courtship indexes were calculated based on 20—30 min observation during a 30 min incubation period.

## Male chaining assays

We inserted newly-eclosed tester males into individual vials, and aged them for 5—7 days. We introduced 8—12 males into a 35 mm Petri dish, which was filled with 8 ml 1.5% agarose through a 2 mm diameter hole drilled on the cover. We allowed the flies to recover for 5 min, and then determined the ratio of time over the next 10 min in which $\geq 3$ flies engaged in simultaneous courtship (chaining index).

## M–M aggression assay

The aggression assays were carried out as described previously (*Zhou et al., 2008*), using 5—7 day-old isolation-housed tester males, and 5—7 days group-housed *w*^1118 males as the targets. Briefly, one tester was paired with one target in the assay. The custom-designed chambers were based on previous reports (*Zhou et al., 2008*; *Liu et al., 2011*), and were fabricated by the Physics Machine Shop at UCSB (*Figure 1—figure supplement 3*). The chamber consists of two concentric circular chambers. The outer chamber diameter and height are 13 mm and 7 mm, respectively. The inner chamber diameter and height are 8 mm and 3.5 mm, respectively. The outer and inner chambers are separated by 0.5 mm thick, 3.5 mm high walls. 0.3 ml standard corn meal and molasses fly food was added to the inner chamber. 1.5% agarose was used to fill the space between inner and outer chambers. The heights of the food and agarose patches were the same (3.5 mm). We then dissolved 15% sucrose and 15% yeast in apple juice, and added 15 μl liquid to each food patch. Once the liquid mixture has soaked into the food, and the patch is dry at the surface, the aggression chamber is ready to use. *w*^1118 male targets were transferred to the chamber by ice-anesthetization. A 22 × 22 mm microscope cover glass (Fisher Scientific) was used to cover to the chamber. The targets were allowed to recover for 10 min, and the isolation-housed tester males were introduced into the chamber by gentle tapping. After waiting 5 min for the tester males to recover, we scored the number of lunges during the following 15 min.

## Male and female preference assay

To test the preference of a male tester for females versus males, we placed one decapitated *w*^1118 virgin female and one decapitated *w*^1118 male in a courtship chamber. The tester males were isolation-housed for 5—7 days since eclosion, and transferred into the chamber by gentle tapping. After 5 min recovery time, we scored the time during which the tester male performed courtship behavior towards either the decapitated female or the decapitated male target over the course of 10 min. The preference index is the ratio of time that male testers spend courting decapitated female targets out of the total courtship time.

## Molecular biology

### Generation of *npf*^1 strain

To generate the *npf*^1 allele (*Figure 1E*) we used the CRISPR mediated NHEJ (clustered regularly interspaced short palindromic repeats – non-homologous end joining) method (*Kondo and Ueda, 2013*; *Ren et al., 2013*).

We designed the following oligonucleotides:

*npf*-gRNA1-f: 5' CTTCGCCCTTGCCCTCCTAGCCGC 3'
*npf*-gRNA1-r: 5' AAACGCGGCTAGGAGGGCAAGGGC 3'
*npf*-gRNA2-f: 5' CTTCGTTGCCATGGTCGTCTAAAA 3'
*npf*-gRNA2-r: 5' AAACTTTTAGACGACCATGGCAAC 3'

We annealed the oligonucleotides to obtain two independent dimers, and ligated the primer dimers into the BbsI site of pU6-BbsI-ChiRNA BbsI (Addgene #45946). The pU6-BbsI-*npf*-gDNA1 and the pU6-BbsI-*npf*-gDNA2 plasmids were co-injected into the BDSC strain #51324 as the Cas9 source (BestGene Plan R). Based on DNA sequencing, we found that *npf¹* harbored a single nucleotide frameshift deletion that changed the 2nd position of codon 19.

## Generation of *npf^LexA* strain

To generate the *npf^LexA* line with an insertion of the *LexA* reporter (**Figure 1D**), we used the CRISPR-HDR (clustered regularly interspaced short palindromic repeats – homology directed repair) method (**Kondo and Ueda, 2013**; **Ren et al., 2013**). We chose upstream and downstream guide RNAs that targeted the *npf* coding sequences using the CRISPR Optimal Target Finder: http://tools.flycrispr.molbio.wisc.edu/targetFinder/.

We annealed the following upstream and downstream primer dimers, which we inserted into the BbsI site of pU6-BbsI-ChiRNA (Addgene #45946).

*npf*_up_ChiRNA_F: 5' CTTCCCAAACAATGCGTTGCATCC 3'
*npf*_up_ChiRNA_R: 5' AAACGGATGCAACGCATTGTTTGG 3'
*npf*_down_ChiRNA_F: 5' CTTCAGATTTATGTTACAGGCTCG 3'
*npf*_down_ChiRNA_R: 5' AAACCGAGCCTGTAACATAAATCT3'

We amplified the *npf* upstream (1359 bp, nucleotides 3 R:16609779 to 16611137, release = r 6.16) and downstream (1388 bp, nucleotides 3 R:16610753 to 16612140, release = r 6.16) homology arms using the following primers:

*npf*_LA_KpnI_F: 5' GACGCATACCAAACGGTACCCATTGTGACACCGTTGCGCTTTCCA 3'
*npf*_LA_KpnI_R: 5' TTTTGATTGCTAGCGAGTTCTATAAATGGCTAATGTATGT 3'
*npf*_RA_NdeI_F: 5' CTAGGCGCGCCCATATGTCGCGGTTTTAATGAGGAGGAGATATTC 3'
*npf*_RA_NdeI_R: 5' GACAAGCCGAACATATGATCGGACTTGGACGTGGTAAGCCAA 3'

We used the In-Fusion cloning kit (Clontech) to clone the upstream and downstream homology arms into the KpnI and NdeI sites of pBP*LexA::p65Uw* (Addgene #26231), respectively.

The pU6-BbsI-ChiRNA-*npf*_up, pU6-BbsI-ChiRNA-*npf*_down, and pBP*LexA::p65Uw*-*npf*_LA + RA plasmids were injected into the BDSC #51323 strain, which provided the source of Cas9 (BestGene Plan R).

We used the following primers to genotype the transformants (as shown in **Figure 1—figure supplement 2A**):

*npf*[*LexA*]LA_GT_F (P1): 5' CTTTCGGCCAACATTTATTCACG 3'
*npf*[*LexA*]LA_GT_R (P2): 5' AAAGCCCAGTCGCTGTGCTATCT 3'
*npf*[*LexA*]RA_GT_F (P3): 5' TCAAATACCCTTGGATCGAAGTA 3'
*npf*[*LexA*]RA_GT_R (P4): 5' AGGGCTGCTGTAAGTATCGGTTG 3'
*npf*_Deletion_F (P5): 5' CTTCCCAAACAATGCGTTGCATCC 3'
*npf*_Deletion_R (P6): 5' AAACCGAGCCTGTAACATAAATCT 3'

## Generation of *npfr^LexA* strain

We employed CRISPR-HDR (**Kondo and Ueda, 2013**; **Ren et al., 2013**) to generate the *npfr^LexA* mutant with the *LexA* knockin. We chose the upstream and a downstream guide RNAs targeting the third exon using the CRISPR Optimal Target Finder.

We annealed the following upstream and downstream primer dimers, which we cloned into the BbsI site of pU6-BbsI-ChiRNA (Addgene #45946).

*npfr*_up_ChiRNA_F: 5' CTTC GCAGATGGGGAGCATCTGAG 3'
*npfr*_up_ChiRNA_R: 5' AAAC CTCAGATGCTCCCCATCTGC 3'
*npfr*_down_ChiRNA_F: 5' CTTC ATTGCGAGCAGTGCGCATGA 3'
*npfr*_down_ChiRNA_R: 5' AAAC TCATGCGCACTGCTCGCAAT 3'

We amplified the *npfr* upstream (1426 bp, nucleotides 3 R:6190969 to 6192394, release = r 6.16) and downstream (1250 bp, nucleotides 3 R:6192051 to 6193300, release = r 6.16) homology arms using the following primers:

*npfr*_LA_KpnI_F: 5' GACGCATACCAAACGGTACC TGTGCTGCATAAATTACGGCGACGG 3'
*npfr*_LA_KpnI_R: 5' TTTTGATTGCTAGCGGTACC AGATGCTCCCCATCTGCCAGCTGGG 3'
*npfr*_RA_NdeI_F: 5' CTAGGCGCGCCCATATG TGCGCACTGCTCGCAATCTGTTCAT 3'
*npfr*_RA_NdeI_R: 5' GACAAGCCGAACATATG CGCGCCCACGAACTGCAGGC 3'

We used the In-Fusion cloning kit (Clontech) to clone the upstream and downstream homology arms into the KpnI and NdeI sites of pBP*LexA::p65Uw* (Addgene #26231).

The pU6-BbsI-ChiRNA-*npf*_up, and pU6-BbsI-ChiRNA-*npf*_down, pBP*LexA::p65Uw-npf*_LA + RA plasmids were co-injected into the BDSC #55821 strain (BestGene Plan R), which provided the source of Cas9.

We used the following primers to genotype the transformants (as shown in *Figure 7—figure supplement 1*):

*npfr*[*LexA*]LA_GT_F (P1): 5' CATGTCTCGCCTTGATGTGCTGC 3'
*npfr*[*LexA*]LA_GT_R (P2): 5' AAAGCCCAGTCGCTGTGCTATCT 3'
*npfr*[*LexA*]RA_GT_F (P3): 5' TCAAATACCCTTGGATCGAAGTAAA 3'
*npfr*[*LexA*]RA_GT_R (P4): 5' CACAGCGAGAAGATCGAGTAGTAGAA 3'

The following primers were used to amplify *npfr* cDNA, which we obtained by performing RT-PCR using mRNA extracted from *npfr*[*LexA*] and control flies:

*npfr*_Deletion_F1 (P5): 5' CACCTCGGATCTGAATGAGACTGG 3'
*npfr*_Deletion_R1 (P6): 5' AGACGATTAGCACGCCGTACATG 3'
*npfr*_Deletion_F2 (P7): 5' CACCCTGGTTGTTATAGCCGTCAT 3'
*npfr*_Deletion_R2 (P8): 5' ACGCACAGCGAGAAGATCGAGTAG 3'

## Generation of P[*g-npf*[+]] transgenic flies

We obtained a plasmid covering 20,306 bp of the *npf* genomic region from P[acman] Resources (http://www.pacmanfly.org/libraries.html). The P[acman] BAC CH322-163E17 plasmid, and a plasmid source of *phiC31* were co-injected into a strain (BDSC #9723) with an *attP40* site (BestGene Plan H).

## Immunohistochemistry

Fly brains were dissected in ice-cold phosphate-buffered saline (PBS, pH 7.4, diluted from a sterile filtered 10x PBS stock, cat#:119-069-131, Quality Biological, Inc. 1x working concentration contains 137 mM NaCl, 2.7 mM KCl, 2 mM $KH_2PO_4$, 8 mM $Na_2HPO_4$) and fixed in 4 % paraformaldehyde in PBST (0.3 % Triton X-100 in PBS) at room temperature for ~ 20 min. Brains were washed three times in PBST for 20 min each time, and blocked in 5 % normal goat serum in PBST for 1 hr. The brains were incubated with primary antibodies diluted in 5 % normal goat serum in PBST for 24 hr at 4 °C. Samples were washed three times with PBST before applying secondary antibodies for 3 hr at 25 °C in darkness. After washing three times with PBST, the samples were mounted with VectaShield (Vector Labs) on glass slides. The primary antibodies were chicken anti-GFP (1:1000, Invitrogen, A-10262), rabbit anti-DsRed (1:1000, Clontech, 632496), mouse nc82 (1:250, Developmental Studies Hybridoma Bank), rabbit anti-FruM (1:10000) (*Stockinger et al., 2005*), rat anti-DsxM (1:500) (*Hempel and Oliver, 2007*) rabbit anti-NPF (1:250 ABIN641365), and mouse anti-V5 (1:500 DyLight549 tagged, MCA2894D549GA BioRad). The secondary antibodies were AlexaFluor 488 goat anti-chicken (1:1000; Invitrogen, A-11039), AlexaFluor 488 goat anti-rat (1:1000; Invitrogen, A-11006), AlexaFluor 568 goat anti-rabbit (1:1000; Invitrogen, A-11011), AlexaFluor 633 goat anti-mouse (1:1000; Invitrogen, A-21050), Rhodamine Red-X goat anti rabbit IgG (1:1000; Molecular Probe, R6394). We adapted a previously described method for anti-V5 and anti-GFP double staining (*Nern et al., 2015*). Briefly, we first used chicken anti-GFP as the primary antibodies (1:1000, Invitrogen, A-10262) for 24 hr 4 °C. We washed the brains three times with PBST, and then added AlexaFluor 488 goat anti-chicken IgG (1:1000; Invitrogen, A-11039) and DyLight549 tagged mouse anti-V5 antibodies (1:500 DyLight549 tagged, MCA2894D549GA BioRad). The brains were incubated at 25 °C for 3 hr in darkness, washed three times in PBST, and mounted with VectaShield (Vector Labs) on glass slides. We performed the imaging using a Zeiss LSM 700 confocal microscope, and processed the images using ImageJ.

## GRASP analysis

To detect native GRASP GFP fluorescence in brains, we used flies aged for ~ 20 days to enhance the reconstructed GFP signals. We dissected the brains in ice-cold PBS, fixed the tissue for 20 min in 4 % paraformaldehyde in PBST at 25 °C, washed three times with PBST, and mounted the brains in PBS for imaging the native fluorescent signals.

## Ex vivo Ca$^{2+}$ imaging

We dissected brains from 7 to 15 day-old males (separated from females for 5 days, raised in ~ 10 male-only group) in cold *Drosophila* imaging saline (108 mM NaCl, 5 mM KCl, 2 mM CaCl$_2$, 8.2 mM MgCl$_2$, 4 mM NaHCO$_3$, 1 mM NaH$_2$PO$_4$5 mM trehalose, 10 mM sucrose, 5 mM HEPES, pH = 7.5 (*Inagaki et al., 2014*), transferred individual brains to 35 mm plastic Petri dishes (35 3001 Falcon), attached the brain down to the bottom of the dish with a slice harp (SHD-26GH/10, Warner Instruments), and bathed each brain in 2 ml *Drosophila* imaging saline. We imaged the Ca$^{2+}$ dynamics using a Zeiss LSM 700 confocal microscope. The images were acquired using a Zeiss 20x water objective (20x/1.0 DIC (uv) VIS-IR, Zeiss) and a 488 nm laser, with the anterior side of the brain facing up to the objective. The images were acquired at a 128 × 128 pixel resolution, and at a frame rate of ~ 10 Hz.~ 10 Z axial sections were imaged in one time-series cycle. The section interval was ~ 1 µm. The time intervals between each cycle were 2 s.

Before stimulating a brain, we imaged the basal GCaMP3 signals for $\geq$ 10 cycles. We then gently added 200 µl 50 mM ATP (pH adjusted to 7.0, Sigma, A2383-5G) into the *Drosophila* imaging saline, resulting in a final ATP concentration of 5 mM. We performed a stack registration using the ImageJ Plugins registration module and measured the GCaMP3 intensities using the ImageJ Analyze ROI manager module. ΔF/F$_0$ (%) was calculated as ΔF/F$_0$ (%)=(F-F$_0$)/F$_0$ × 100. F$_{max}$ is the maximum fluorescence value following ATP delivery. F$_{min}$ is the minimum fluorescence value that occurred during a total of 80 time series cycles after ATP delivery. F$_0$ is the GCaMP3 baseline value averaged for 10 time-series cycles immediately before ATP application.

## Statistical analyses

No statistical methods were employed to predetermine sample sizes. Sample sizes were chosen based on previous publications (*Demir and Dickson, 2005*; *Manoli et al., 2005*; *Stockinger et al., 2005*; *Pan et al., 2012*; *Asahina et al., 2014*; *Clowney et al., 2015*; *Huang et al., 2016*; *Zhang et al., 2016*). Statistical analysis was performed with Prism5 (GraphPad Software). We performed nonparametric Mann-Whitney test when comparing two groups of data. For comparison of multiple groups of data, we performed Kruskal-Wallis test followed by Dunn's *post hoc* test. * indicates $p < 0.05$, ** indicates $p < 0.01$, *** indicates $p < 0.001$. We present the exact number of samples and *P* values in the figure legends and in the supporting source data files. We present raw data using scatter plots and include exact values in the source data files. When n < 10, individual data points were identified.

## Replication

We used only biological replicates throughout this work. To perform the behavioral studies, we defined biological replicates as animals of the same genotype and rearing conditions, exposed to identical treatments. Courtship indexes were calculated using n = 6—27 individual animals. Preference indexes were calculated using n = 12 individual animals. Chaining indexes were calculated using n = 6 groups (8—12 individual animals in each group). Lunging numbers were calculated using n = 10—12 animals. All animals were used once, since their behavioral indexes are sensitive to prior experience. Replicates for the Ca$^{2+}$ imaging were defined as the number of neurons (*Figure 6C and F*) or the selected regions (*Figure 8E*) analyzed per genotype and condition. In all cases we used 3—9 brains/genotype and condition. 2—5 neurons (*Figure 6C and F*) or 4—7 regions of selection (*Figure 8E*) were used per brain. Replicates for the immunostaining were defined as brains of the same genotype that underwent identical staining procedures. We stained $\geq$ 5 brains per experiment. The *Gal4/UAS* (or *LexA/LexAop*) binary systems are highly reproducible. Images that were the most intact were selected for display. We did not exclude any data points.

## Group allocation

To perform the behavioral assays, the control and experimental groups were reared under the same conditions, collected on the same day, aged in parallel, and assayed on the same day. The control and experimental groups were assayed in an arbitrary order. Behavioral videos were randomly permuted for scoring behavioral indexes. All behavioral analyses were obtained from videos, in which the genotypes were masked. The indexes were calculated blindly.

To perform the $Ca^{2+}$ imaging, the control and experimental groups were assayed in an arbitrary order. The raw $Ca^{2+}$ imaging data files were permutated in order and analyzed by Image J software.

## Source data files

The raw data for the behavioral assays, $Ca^{2+}$ imaging assays, summary statistics, and full stacks of the entire brains used in the GRASP experiments are included in the source data files.

## Acknowledgements

This work was supported by grants to CM from the National Institute on Deafness and other Communication Disorders (DC007864) and an NIH Director's Pioneer Award (DP1) supported by the National Institute of Allergy and Infectious Disease (DP1AI124453). We thank Drs. Barry Dickson (Janelia Research Campus), Ping Shen (University of Georgia), Gerald Rubin (Janelia Research Campus), Christopher Potter (Johns Hopkins University School of Medicine), Brian Oliver (NIDDK, NIH), Orie Shafer (University of Michigan), David Anderson (California Institute of Technology) for sharing valuable fly stocks and experimental reagents. We thank Drs. Julie Simpson, Hsiang-Chin Chen, Junjie Luo and Jiangqu Liu (UC Santa Barbara) for critical discussions and technical advice and the Physics Machine Shop (UC Santa Barbara) for fabricating behavioral chambers. We thank Dr. Wen Wang and Ruoping Zhao in the Kunming Institute of Zoology, Chinese Academy of Sciences for providing lab resources during the author's job transition, which greatly facilitated the completion of this work.

## Additional information

### Funding

| Funder | Grant reference number | Author |
|--------|------------------------|--------|
| National Institute on Deafness and Other Communication Disorders | DC007864 | Craig Montell |
| National Institute of Allergy and Infectious Diseases | DP1AI124453 | Craig Montell |

The funders had no role in study design, data collection and interpretation, or the decision to submit the work for publication.

### Author contributions

Weiwei Liu, Conceptualization, Resources, Data curation, Formal analysis, Investigation, Visualization, Methodology, Writing—original draft; Anindya Ganguly, Conceptualization, Formal analysis, Investigation, Writing—review and editing; Jia Huang, Resources, Investigation, Methodology; Yijin Wang, Investigation, Methodology, Writing—review and editing; Jinfei D Ni, Resources, Investigation; Adishthi S Gurav, Morris A Aguilar, Validation, Investigation; Craig Montell, Conceptualization, Formal analysis, Supervision, Funding acquisition, Writing—original draft, Project administration

### Author ORCIDs

Weiwei Liu (iD) https://orcid.org/0000-0001-5082-9114
Jia Huang (iD) http://orcid.org/0000-0001-8336-1562
Yijin Wang (iD) https://orcid.org/0000-0001-5488-3089
Jinfei D Ni (iD) http://orcid.org/0000-0002-7004-1241

Craig Montell https://orcid.org/0000-0001-5637-1482

**Decision letter and Author response**
Decision letter https://doi.org/10.7554/eLife.49574.058
Author response https://doi.org/10.7554/eLife.49574.059

## Additional files

**Supplementary files**
• Supplementary file 1. Key Resources Table.
DOI: https://doi.org/10.7554/eLife.49574.054

• Supplementary file 2. Genotypes of flies used in each figure and video.
DOI: https://doi.org/10.7554/eLife.49574.055

• Transparent reporting form
DOI: https://doi.org/10.7554/eLife.49574.056

**Data availability**

All data generated or analysed during this study are included in the manuscript and supporting files.

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
