## [Decision Letter]

[Editors’ note: a previous version of this study was rejected after peer review, but the authors submitted for reconsideration. The first decision letter after peer review is shown below.]

Thank you for submitting your work entitled "Male-specific neuropeptide F neurons control *Drosophila* male sex drive through recurrent inhibition" for consideration by *eLife*. Your article has been reviewed by two peer reviewers, and the evaluation has been overseen by Mani Ramaswami as Reviewing Editor and a Senior Editor. The reviewers have opted to remain anonymous.

Our decision has been reached after consultation between the reviewers. Based on these discussions and the individual reviews below, we regret to inform you that your work will not be considered further for publication in *eLife*. While the findings are potentially interesting and appropriate for *eLife*. There are several experiments required to address current weaknesses in the manuscript, which the reviewers agreed would take longer than the period allowed for revision. It is *eLife*'s policy to not invite revisions that require extensive new experiments so that authors are free to submit elsewhere if they disagree with our assessment. If, however, you agree that the proposed experiments would strengthen the paper and you wish to do them, *eLife* would be glad to consider a completely revised manuscript as a new submission.

In case it is useful, we provide below a consensus review, as well as the original reviews of your manuscript.

Summary:

In this manuscript, the authors identify a group of (fruM+) male-specific neuropeptide F (*NPF^M^*) neurons in *Drosophila* and provide evidence to show that these participate an inhibitory feedback circuit with a group of courtship command neurons called P1 through a third group of NPF-Receptor expressing neurons. Activation of P1 cells results in activation of NPF neurons, which inhibit NPF-receptor neurons that normally activate P1. Thus, the data argue for a recurrent/ feedback inhibitory circuit that modulates P1. They further show that NPF neurons are essential for reducing levels of courtship, by showing that loss of NPF in *NPF^M^* neurons results in increased courtship and mating. They conclude from these data that this circuit is important for regulating male sex drive and/or satiety.

One major weakness of this manuscript is the lack of neuronal specificity in most experiments, which undermines that authors' claim that the all of the behavioral, physiological and anatomical data provided reflect interactions of the fruM+ subset of NPF-expressing NPF^M^ neurons (as against other NPF neurons) with male courtship-promoting P1 neurons (as against other P1 neurons). Indeed, the only direct data linking NPF^M^ neurons and courtship behavior is the RNAi-mediated knockdown of the npf gene in fruM+ neurons using NP21 (Supplementary Figure 4). Additional experiments are required to address this issue of which some possibilities are outlined later in this critique.

A second conceptual issue is that while this circuit modulates courtship and mating, there is no conclusive evidence that it underlies sexual drive/sexual satiation per se, as suggested by the authors. If the authors want to make this claim, additional evidence would be necessary to support this conclusion, such as calcium imaging experiments showing that sexual satiety or deprivation alters the activity of NPF neurons, and that P1 activation plays an essential role in this process. It would also support their conclusions if they showed that the time course of activation of these neurons after sexual activity corresponds to that of persistence of sexual satiation. However, as experiments are beyond the scope of the current study, it would be sufficient to alter the text and conclusions in this regard rather than adding the additional experiments.

Essential revisions:

It is essential that the issue of cell-type specificity be addressed. Possible experimental approaches (and rationales for why these experiments are required) are listed below.

1) Because NPF npf-GAL4 and npf^LexA^ labels many non-fruM+ neurons (Figure 2), additional experiments are needed to show that increase of courtship by the blockade of synaptic transmission in npf-GAL4 neurons (Figure 1A-C) is accounted for by NPF^M^ neurons. At minimum the effect of selective NPF^M^ neuron activation (TrpA1 or ChR) and inhibition (possibly Shi^ts^) or courtship and mating behavior should be examined.

2) For the same reason, it is not clear whether the GRASP signal (Figure 4G,H), and functional interactions (GCaMP) between npf^LexA^ and R71G01-GAL4 ("P1-GAL4") (Figure 5A-C), are indeed generated by NPF^M neurons and P1 neurons that control male courtship. Considering the widespread arborization of npf^LexA^ neurons, it is entirely possible that the non-fruM+ population of npf^LexA^ neurons affects GRASP and Ca^2+^ signals. Likewise, the R71G01 promoter does not specifically label P1 neurons. This is a particular concern in Figure 5D-F because the activation of npf-GAL4 neurons can be caused by non-P1 neurons. Lastly, it is not clear how authors can unambiguously determine that the neurons exhibiting Ca^2+^ signals in Figure 5 and Figure 7 are NPF^M^ or P1 neurons, when the GAL4 drivers are not specific.

The most straightforward way to address this weakness is to conduct all experiments using UAS or LexAop transgenes with genetic intersection approach that combines a "FRT-stop-FRT" cassette and fru^FLP^. While the authors may not be able to truly claim their conclusion without specific manipulation of NPF^M^ neurons, we suggest a possible set of additional of experiments (including 1 above) that could help effectively alleviate specificity concerns. However, we acknowledge there could be alternative experiments or arguments to address this issue.

2a) Related to (1) above. Using existing "FRT-stop-FRT" transgenic elements (related to Figure 1 and Figure 1—figure supplement 1): The "FRT-stop-FRT" versions of shibire^ts^ and tetanus toxin light chain (TNT) have been previously published (Stockinger et al., 2005), and can be combined with fru^FLP^ to selectively impair function of NPF^M^ neurons. Authors can perform the courtship assay using these reagents to address whether functional impairment of NPF^M^ neurons results in increased courtship.

2b) Using tub-FRT-GAL80-FRT-stop transgenic elements (related to Figure 1, Figure 4 and Figure 5): Authors can block the activity of npf-GAL4 except in NPF^M neurons by combining the tub-FRT-GAL80-FRT transgene (Gordon and Scott, 2009; Bohm et al., 2010) and fru^FLP^. This will allow authors to address if the observed behavioral phenotype using UAS-shibire^ts^ (Figure 1A-C, Figure 4I-K), or observed increase in Ca^2+^ signals (Figure 5D-F), is indeed caused by NPF^M^ neurons.

2c) Using P1a split GAL4: The split GAL4 (15A01-AD; 71G01-DBD: Hoopfer et al., 2015) labels only P1 neurons, ensuring high degree of specificity for GRASP and Ca^2+^ signals, allowing authors to determine if the increase of Ca^2+^ in NPF^M^ neurons is indeed caused by the P1 neurons.

*Reviewer #2:*

In this manuscript, the authors identify a group of male-specific neuropeptide F (NPF) neurons in *Drosophila*, which form an inhibitory feedback circuit with a group of courtship command neurons called P1. They further show that NPF neurons are essential for maintaining normal levels of courtship. They conclude that this circuit is important for signalling the male's sex drive.

The evidence for the logic of the proposed circuit is convincing and well presented. However, there is no conclusive evidence that this circuit underlies sexual drive/sexual satiation as suggested by the authors. If the authors want to make this claim, additional evidence would be necessary to support this conclusion, such as calcium imaging experiments showing that sexual satiety or deprivation alters the activity of NPF neurons, and that P1 activation plays an essential role in this process. It would also support their conclusions if they showed that the time course of activation of these neurons after sexual activity corresponds to that of persistence of sexual satiation. However, these experiments might be beyond the scope of the current study. We would therefore suggest altering the conclusions rather than adding the additional experiments.

*Reviewer #3:*

This paper investigates an important question concerning the mechanism by which neuromodulation affects courtship. To this end, authors created novel genetic reagents to manipulate the neuropeptide F (NPF) system, which is a technical accomplishment that benefits a wide range of *Drosophila* researchers. However, the lack of neuronal specificity throughout the experiments is a major weakness of this manuscript, undermining authors' main claim that the fruM+ population of NPF-expressing neurons (NPF^M neurons) and courtship-promoting P1 neurons form a reciprocal network to regulate courtship. I think the manuscript will be of general interest to the field of neuroscience and genetics only if this weakness is properly addressed.

While authors conclude that npf-GAL4 and npf^LexA^ neurons affect behavior and physiology through NPF^M neurons, no experiments were conducted that selectively manipulated NPF^M neurons. Both npf-GAL4 and npf^LexA^ labels many non-fruM+ neurons (Figure 2). The only piece of data that links NPF^M neurons and courtship behavior is the RNAi-mediated knockdown of the npf gene in fruM+ neurons using NP21 (Supplementary Figure 4). However, these data do not prove that increase of courtship by the blockade of synaptic transmission in npf-GAL4 neurons (Figure 1A-C) is accounted for by NPF^M^ neurons. Neither does it mean that GRASP signal (Figure 4G,H), nor the functional interactions between npf^LexA^ and R71G01-GAL4 ("P1-GAL4") (Figure 5A-C), is indeed generated by NPF^M neurons and P1 neurons. Considering the widespread arborization of npf^LexA^ neurons, it is entirely possible that the non-fruM+ population of npf^LexA^ neurons affects GRASP and Ca^2+^ signals. Likewise, the R71G01 promoter does not specifically label P1 neurons. This is a particular concern in Figure 5D-F because the activation of npf-GAL4 neurons can be caused by non-P1 neurons. Lastly, I am unsure how authors can unambiguously determine that the neurons exhibiting Ca^2+^ signals in Figure 5 and Figure 7 are NPF^M^ or P1 neurons, when the GAL4 drivers are not specific.

The most straightforward way to address this weakness is to conduct all experiments using UAS or LexAop transgenes with genetic intersection approach that combines a "FRT-stop-FRT" cassette and fru^FLP, which may not be readily available to the authors. While I do not think the authors can truly claim their conclusion without specific manipulation of NPF^M neurons, I think a following set of experiments will effectively address specificity concerns:

[Editors’ note: what now follows is the decision letter after the authors submitted for further consideration.]

Thank you for submitting your article "Male-specific neuropeptide F neurons regulate courtship in *Drosophila* by recurrent inhibition of P1 neurons" for consideration by *eLife*. Your article has been reviewed by two peer reviewers, including Mani Ramaswami as the Reviewing Editor and Reviewer #1, and the evaluation has been overseen by Eve Marder as the Senior Editor.

The reviewers have discussed the reviews with one another and the Reviewing Editor has drafted this decision to help you prepare a revised submission.

Summary:

This extensively revised new manuscript by Liu et al., addresses many of the major concerns raised for the previous submission. The data presented provided strong support for the authors' conclusion that the fru-expressing subpopulation of NPF-releasing neurons (hereafter shorthanded as NPF^M^ neurons), as well as NPF itself, act to suppress courtship behavior. The connections from the P1 neurons to NPF^M^ neurons are also convincingly demonstrated. The authors are commended and congratulated for the effort put into this work.

However, there remain three concerns that need to be addressed, at least by revisions to the interpretations in the text and by providing additional raw data that will allow independent and different analyses.,

Essential revisions:

1) Interpretation of the activation of NPF^M^ neurons:

The authors argue that the artificial activation of NPF^M^ neurons (Figure 4B) decreases courtship, in contrast to the silencing of the same neurons (Figure 4A, also Figure 1A-C, Figure 1—figure supplement 1A). This conclusion is solely based on their observation that the courtship index (CI) of the experimental genotype (right most) decreased in 31 degree compared to in 23 degree. However, plots suggest that the CI of the experimental genotype at 31 degree may not be significantly different from genetic controls in the same temperature. In fact, the statistical significance between 2 temperatures of the experimental genotype seems largely due to the higher level of courtship behaviors of this genotype at 23 degree compared to the controls. Low number of n for "no fru^FLP^" control (only 8 for 31 degree) is also a concern. The authors do not provide statistical tests across genotypes or source data for readers to perform tests on their own. However, it is important to conduct inter-genotype comparison because authors rely on this piece of data to argue the instructive (not permissive) courtship-suppressing role of NPF^M^ neurons.

In summary, while the authors perform multiple manipulations to convincingly demonstrate that the NPF^M^ neurons and NPF from these neurons are necessary to maintain the proper levels of courtship. it remains a stretch to claim that the activation of NPF^M^ neuron decreases courtship without support from inter-genotype comparisons at 31C. Either these comparisons should be done and shown to support the conclusion, or the conclusions should be duly modulated.

This issue also raises caveats relevant to the recurrent inhibitory loop model (see below).

2) On the "recurrent inhibitory loop" model.

A major conceptual conclusion of this manuscript is the "recurrent inhibitory loop" model presented in Figure 8F. However, more data are needed to clearly demonstrate that NPF^M^ neurons feed back to P1 neurons for the following reasons. First, and most importantly, only data that directly shows the "inhibitory" role of NPF^M^ neurons (behaviorally or physiologically) is the Figure 4B (inhibition of courtship behavior), the problem of which I already discussed above. No data support that NPF^M^ neurons inhibit the NPFR-expressing putative downstream neurons.

In addition, a specific subset among NPFR-expressing neurons involved in courtship modulation was not identified in this study (all experiments involving NFPR-expressing neurons were done with NPFR^LexA^, which labels many neurons as shown in Figure 7E). It is therefore unclear whether NPFR-expressing downstream of NPF^M^ neurons and the NPFR-expressing neurons that activate P1 neurons (Figure 8A-E) are the same population. The only remaining data in support of the authors' claim that NPF^M^ neurons act through P1 neurons are results of behavioral epistasis experiments in Figure 7A and B. However, it is difficult to interpret these data because hi^ts^-mediated silencing of P1 neuronal transmission alone does not seem to reduce courtship behavior – which one would expect. While authors' data strongly support that P1 neurons activate NPF^M^ neurons, it remains fully possible that NFPM neurons do not make a "recurrent" connection with P1 neurons. Instead, a common downstream neuron may integrate activities of NPF^M^ and P1 neurons, as authors acknowledge as an alternative mechanism (subsection “NPF^M^neurons inhibits P1 neurons indirectly through a feedback loop”).

Given these missing experimental/ logical elements, it seems that the model in Figure 8F is not the only reasonable mechanism to account for their observations. There needs to be a revised and more conservative discussion of these data. The "recurrent inhibitory loop" model, should only be presented as one possible mechanism with clear acknowledgement of alternative possibilities and outstanding issues that need to be resolved.

3) Source data for behavioral and histology experiments.

3A) Raw data for behavioral experiments should be included. This is especially important for the issue regarding the data in Figure 4B raised in the major comments. Without the raw data, it is impossible to statistically re-analyze the data. Perhaps it would also be useful to show statistical values relevant for non-normal distributions (alternatives to the mean, standard deviation and standard error), given that the statistics used involve the non-parametric Mann-Whitney U test, which concerns median.

3B) Also, GRASP experiments (Figure 5A-C, G-I, Figure 8A, B) are not informative unless (1) examples of negative controls, most importantly without a driver for GFP11 fragments), and (2) description of z-stack, are present. Authors use female brains as a proxy for negative controls, but it is not a fair comparison given that the expression of GFP1-10 (under the control of R71G01-GAL4) are likely different between the two sexes. Also, it is not clear whether the presented whole brain images are z-stack across the entire brain or a partial z-stack. The full z-stack is necessary to address whether the GRASP signal they presented in the panels above are the only signals they observed across the brain.

---

## [Author Response]

[Editors’ note: the author responses to the first round of peer review follow.]

Summary:[…] One major weakness of this manuscript is the lack of neuronal specificity in most experiments, which undermines that authors' claim that the all of the behavioral, physiological and anatomical data provided reflect interactions of the fruM+ subset of NPF-expressing NPF^M^ neurons (as against other NPF neurons) with male courtship-promoting P1 neurons (as against other P1 neurons). Indeed, the only direct data linking NPF^M^ neurons and courtship behavior is the RNAi-mediated knockdown of the npf gene in fruM+ neurons using NP21 (Supplementary Figure 4). Additional experiments are required to address this issue of which some possibilities are outlined later in this critique.

We added new experiments to resolve the neuronal specificity concern, which are described in response to the specific comments in the Essential revisions.

A second conceptual issue is that while this circuit modulates courtship and mating, there is no conclusive evidence that it underlies sexual drive/sexual satiation per se, as suggested by the authors. If the authors want to make this claim, additional evidence would be necessary to support this conclusion, such as calcium imaging experiments showing that sexual satiety or deprivation alters the activity of NPF neurons, and that P1 activation plays an essential role in this process. It would also support their conclusions if they showed that the time course of activation of these neurons after sexual activity corresponds to that of persistence of sexual satiation. However, as experiments are beyond the scope of the current study, it would be sufficient to alter the text and conclusions in this regard rather than adding the additional experiments.

We altered the Title and text accordingly.

Essential revisions:It is essential that the issue of cell-type specificity be addressed. Possible experimental approaches (and rationales for why these experiments are required) are listed below.1) Because NPF npf-GAL4 and npf^LexA^ labels many non-fruM+ neurons (Figure 2), additional experiments are needed to show that increase of courtship by the blockade of synaptic transmission in npf-GAL4 neurons (Figure 1A-C) is accounted for by NPF^M^ neurons. At minimum the effect of selective NPF^M^ neuron activation (TrpA1 or ChR) and inhibition (possibly Shi^ts^) or courtship and mating behavior should be examined.

To address the concern regarding the specificity of NPF^M^ neurons in regulating male courtship, we added new experiments using FlpOut methods to specifically silence or activate NPF^M^ neurons respectively. In addition, we introduced the Gal4 repressor, Gal80, in either NPF^M^ neurons or *fru^-^* NPF neurons to confine *UAS-npf*RNAi expression to *fru^-^* NPF neurons or NPF^M^ neurons respectively.

Our new results are as follows:

A) Specific disruption of NPF^M^ neurons

We assayed male courtship in *npf-Gal4/UAS>stop>Shibire^ts^;fru^FLP^*/+ flies. In the *npf* and *fru* double positive neurons, Gal4 and Flippase and are expressed.

Consequently, the “*stop*” cassette is excised, and Shibire^ts^ is expressed specifically in NPF^M^ neurons. When we assayed flies at the restrictive temperature for Shi^ts^ (31°C), the males exhibited elevated M–F courtship (Figure 4A, subsection “Sexual dimorphic NPF^M^neurons suppress male courtship”).

We then tested the effects of inhibiting neurons except for NPF^M^ neurons, using *npfGal4/UAS>Shi^ts^>stop;fru^FLP^*/+ flies, which removes *Shi^ts^* exclusively in NPF^M^ neurons. These males displayed similar levels of courtship at both the permissive temperature and non-permissive temperatures for Shi^ts^ (Figure 4A, subsection “Sexual dimorphic NPFM neurons suppress male courtship”).

B) Specific activation of NPF^M^ neurons

To activate NPF^M^ neurons, we employed a similar FlpOut approach, using

*UAS>stop>trpA1/npf-Gal4;fru^FLP^*/+ flies, to express the thermally-activated TRPA1-A isoform in NPF^M^ neurons only. The TRPA1-A isoform is a Na+ and Ca^2+^-permeable channel, which is gated at temperatures above ~27°C (Viswanathet al.,, 2003). To perform these assays, we used decapitated females since intact females induce ceiling levels of male courtship, which are resistant to downregulation, while decapitated females induce moderate levels, which facilitate detecting subtle decreases in male courtship. We found that courtship levels in *UAS>stop>trpA1/npf-Gal4; fru^FLP^/*+ males were suppressed at 29° relative to 23°C (Figure 4B). In contrast, none of the three types of control flies exhibited lower male courtship at 29°C (Figure 4B). These results (described in subsection “Sexual dimorphic NPF^M^neurons suppress male courtship”) indicate that sexually dimorphic NPF^M^ neurons inhibit male courtship.

C) Specific UAS-npf-RNAi knock down in NPF^M^ neurons

We previously provided evidence that NPF produced in NPF^M^ neurons is responsible for inhibiting male courtship, by knocking down NPF expression in distinct groups of neurons. To conduct these experiments, we used *UAS-npf-RNAi*, which was effective as it greatly reduced NPF levels (Figure 4C-E). Knocking down *npf* expression with the *fru-Gal4* induced a dramatic increase in M–M courtship, and did so to a similar extent as when we employed a pan-neuronal (*elav) Gal4* or the *npf-Gal4* (Figure 4F).

To specifically interrogate a requirement for NPF in NPF^M^ neurons, we used the FlpOut method to introduce *Gal80* (which binds and inhibits *Gal4* activity) in either *fru*^+^ or *fru*^-^ neurons, thereby confining *UAS-npf-RNAi* expression to *fru*^-^ NPF neurons or NPF^M^ neurons, respectively. To knockdown *npf* specifically in NPF^M^ neurons, we used the following flies that causes excision of *Gal80* in *fru* neurons only, thereby allowing *Gal4* expression and RNA knockdown in NPF^M^ neurons: *npfGal4/tub>Gal80>stop;UAS-npf-RNAi/fru^FLP^* flies. Conversely, to prevent *npf* knockdown in NPF^M^ neurons, we expressed *Gal80* specifically in these neurons by taking advantage of *npf-Gal4/tub>stop>Gal80;UAS-npf-RNAi/fru^FLP^* flies. We found that knocking down *npf* exclusively in NPF^M^ neurons elevated M–M courtship while *npf* knock down in *fru*^-^ NPF neurons did not change the level of male courtship (Figure 4F). These results (described in subsection “Sexual dimorphic NPF^M^neurons suppress male courtship”) indicate that NPF produced in sexually dimorphic NPF^M^ neurons are exclusively required to suppress male courtship.

2) For the same reason, it is not clear whether the GRASP signal (Figure 4G,H), and functional interactions (GCaMP) between npf^LexA^ and R71G01-GAL4 ("P1-GAL4") (Figure 5A-C), are indeed generated by NPF^M neurons and P1 neurons that control male courtship. Considering the widespread arborization of npf^LexA^ neurons, it is entirely possible that the non-fruM+ population of npf^LexA^ neurons affects GRASP and Ca^2+^ signals. Likewise, the R71G01 promoter does not specifically label P1 neurons. This is a particular concern in Figure 5D-F because the activation of npf-GAL4 neurons can be caused by non-P1 neurons. Lastly, it is not clear how authors can unambiguously determine that the neurons exhibiting Ca^2+^ signals in Figure 5 and Figure 7 are NPF^M^ or P1 neurons, when the GAL4 drivers are not specific.The most straightforward way to address this weakness is to conduct all experiments using UAS or LexAop transgenes with genetic intersection approach that combines a "FRT-stop-FRT" cassette and fru^FLP^. While the authors may not be able to truly claim their conclusion without specific manipulation of NPF^M^ neurons, we suggest a possible set of additional of experiments (including 1 above) that could help effectively alleviate specificity concerns. However, we acknowledge there could be alternative experiments or arguments to address this issue.

NPF^M^ neurons can be differentiated from other NPF neurons based on their position in the anterior brain region that is immediately adjacent to antennal lobe. Moreover, NPF^M^ form a cluster of 3-5 neurons and their cell bodies are smaller than the pair of dorsal medial and the pair of dorsal lateral large NPF neurons (refer to Figure 2 and Figure 3E-G).

The GRASP GFP signals created by expression of the split GFP using the *npf^LexA^* and *R71G01-GAL4* appear to be due to expression of the two parts of the split GFP in NPF^M^ and P1 neurons for the following reasons. First, NPF^M^ and P1 neurons are both male-specific, and the GRASP signals are primarily in the male brain. Second, the GRASP signals label two LPC structures: the lateral junction and the SMPr arch (Figure 5G-I). Third, the projections of NPF^M^ and P1 neurons overlap extensively in the lateral junction and SMPr arch (Figure 5D-F and Video 2), while *fru^-^* NPF projections do not stain the LPC (Figure 3E-G and Video 1). Thus, the GRASP signals in the LPC structure appear to be formed by interactions between NPF^M^ and P1 neurons. We added a discussion of the specificity of the GRASP signals in the revised manuscript (subsection “P1 neurons directly activate NPF^M^neurons”).

2a) Related to (1) above. Using existing "FRT-stop-FRT" transgenic elements (related to Figure 1 and Figure 1—figure supplement 1): The "FRT-stop-FRT" versions of shibir^ts^ and tetanus toxin light chain (TNT) have been previously published (Stockinger et al., 2005), and can be combined with fru^FLP^ to selectively impair function of NPF^M neurons. Authors can perform the courtship assay using these reagents to address whether functional impairment of NPF^M^ neurons results in increased courtship.

We conducted the suggested experiments, and describe the results in our reply to question 1. Briefly, we combined *fru^FLP^* with *UAS>stop>shibire^ts^* or *UAS>stop>trpA1* to specifically silence or activate NPF^M^ neurons respectively (Figure 4A and B). The experimental results support a specific role of NPF^M^ neurons in suppressing male courtship.

2b) Using tub-FRT-GAL80-FRT-stop transgenic elements (related to Figure 1, Figure 4 and Figure 5): Authors can block the activity of npf-GAL4 except in NPF^M neurons by combining the tub-FRT-GAL80-FRT transgene (Gordon and Scott, 2009; Bohm et al., 2010) and fru^FLP^. This will allow authors to address if the observed behavioral phenotype using UAS-shibire^ts^ (Figure 1A-C, Figure 4I-K), or observed increase in Ca^2+^ signals (Figure 5D-F), is indeed caused by NPF^M^ neurons.

We performed behavioral experiments using the Gal80, and describe the results in our reply to question 1. Briefly, we introduced Gal80 in either NPF^M^ neurons *or fru^-^* NPF neurons to specifically knockdown *npf* expression in *fru^-^* NPF neurons or NPF^M^ neurons respectively. We found that there was an increase in courtship when we knocked down *npf* in NPF^M^ neurons, but not when we knocked down *npf* in *fru^-^* NPF neurons (Figure 4F). These findings also support the conclusion that NPF produced in NPF^M^ neurons serves to suppress male courtship.

2c) Using P1a split GAL4: The split GAL4 (15A01-AD; 71G01-DBD: Hoopfer et al., 2015) labels only P1 neurons, ensuring high degree of specificity for GRASP and Ca^2+^ signals, allowing authors to determine if the increase of Ca^2+^ in NPF^M^ neurons is indeed caused by the P1 neurons.

As suggested, in order to investigate if elevated Ca^2+^ signals in NPF^M^ neurons are caused by specific activation of P1 neurons, we generated flies with the following genotype: *UAS-P2X2,LexAop-GCaMP3/R15A01-AD;npf^LexA^/R71G01-DBD*. The combination of *R15A01-AD* (transcription activation domain) and *R71G01-DBD* (Gal4 DNA binding domain) generates a specific *P1-Gal4* driver to express *UAS-P2X2*. The *npf^LexA^*directs expression of *LexAop-GCaMP3*. Application of ATP activates the *P2X2* receptor, leading to stimulation of P1 neurons. We then monitored the activity of NPF^M^ neurons using GCaMP. We detected significant increases in Ca^2+^ signals in NPF^M^ neurons when we applied ATP, which specifically activates P1 neurons (Figure 6D-F). These data further support the conclusion that activation of P1 neurons, in turn activates NPF^M^ neurons.

[Editors' note: the author responses to the re-review follow.]

Summary:[…] Essential revisions:1) Interpretation of the activation of NPF^M^ neurons:The authors argue that the artificial activation of NPF^M^ neurons (Figure 4B) decreases courtship, in contrast to the silencing of the same neurons (Figure 4A, also Figure 1A-C, Figure 1—figure supplement 1A). This conclusion is solely based on their observation that the courtship index (CI) of the experimental genotype (right most) decreased in 31 degree compared to in 23 degree. However, plots suggest that the CI of the experimental genotype at 31 degree may not be significantly different from genetic controls in the same temperature. In fact, the statistical significance between 2 temperatures of the experimental genotype seems largely due to the higher level of courtship behaviors of this genotype at 23 degree compared to the controls. Low number of n for "no fru^FLP^" control (only 8 for 31 degree) is also a concern. The authors do not provide statistical tests across genotypes or source data for readers to perform tests on their own. However, it is important to conduct inter-genotype comparison because authors rely on this piece of data to argue the instructive (not permissive) courtship-suppressing role of NPF^M^ neurons.In summary, while the authors perform multiple manipulations to convincingly demonstrate that the NPF^M^ neurons and NPF from these neurons are necessary to maintain the proper levels of courtship. it remains a stretch to claim that the activation of NPF^M^ neuron decreases courtship without support from inter-genotype comparisons at 31C. Either these comparisons should be done and shown to support the conclusion, or the conclusions should be duly modulated.This issue also raises caveats relevant to the recurrent inhibitory loop model (see below).

We performed Kruskal-Wallis tests to compare courtship indexes (CIs) across genotypes at 29°C and found that were no significant differences between *UAS>stop>trpA1/npf-Gal4;fruFLP*/+ males, and any of the control males. Therefore, even though there is a significant difference between *UAS>stop>trpA1/npfGal4;fruFLP*/+ males at 23°C and 29°C, we agree that the data preclude us from drawing the conclusion activation of NPF^M^ neurons suppresses male courtship. This point is now made as follows in the Results section: “Nevertheless, because the CI exhibited by the *UAS>stop>trpA1/npf-Gal4;fru^FLP^*/+males at 29°C was not elevated relative to the CIs displayed by the control males at 29°C, the results preclude the conclusion that activation of sexually dimorphic NPF^M^ neurons inhibits male courtship.” In addition, we now provide the raw data for Figure 4B and other behavioral assays in source data files.

2) On the "recurrent inhibitory loop" model.A major conceptual conclusion of this manuscript is the "recurrent inhibitory loop" model presented in Figure 8F. However, more data are needed to clearly demonstrate that NPF^M^ neurons feed back to P1 neurons for the following reasons. First, and most importantly, only data that directly shows the "inhibitory" role of NPF^M^ neurons (behaviorally or physiologically) is the Figure 4B (inhibition of courtship behavior), the problem of which I already discussed above. No data support that NPF^M^ neurons inhibit the NPFR-expressing putative downstream neurons.In addition, a specific subset among NPFR-expressing neurons involved in courtship modulation was not identified in this study (all experiments involving NFPR-expressing neurons were done with NPFR^LexA^, which labels many neurons as shown in Figure 7E). It is therefore unclear whether NPFR-expressing downstream of NPFM neurons and the NPFR-expressing neurons that activate P1 neurons (Figure 8A-E) are the same population. The only remaining data in support of the authors' claim that NPF^M^ neurons act through P1 neurons are results of behavioral epistasis experiments in Figure 7A and B. However, it is difficult to interpret these data because Shi^ts^-mediated silencing of P1 neuronal transmission alone does not seem to reduce courtship behavior – which one would expect. While authors' data strongly support that P1 neurons activate NPF^M^ neurons, it remains fully possible that NPF^M^ neurons do not make a "recurrent" connection with P1 neurons. Instead, a common downstream neuron may integrate activities of NPF^M^ and P1 neurons, as authors acknowledge as an alternative mechanism (subsection “NPF^M^neurons inhibits P1 neurons indirectly through a feedback loop”).Given these missing experimental/ logical elements, it seems that the model in Figure 8F is not the only reasonable mechanism to account for their observations. There needs to be a revised and more conservative discussion of these data. The "recurrent inhibitory loop" model, should only be presented as one possible mechanism with clear acknowledgement of alternative possibilities and outstanding issues that need to be resolved.

We revised the manuscript in all of the relevant sections so as to be more conservative concerning the “the recurrent inhibitory loop” model. We no longer highlight this model in the Title, Abstract, and at the end of the Introduction. Most importantly, we expanded the Discussion section and added Figure 8G to clarify the limitations in stating that the recurrent inhibitory loop is the only model. We specifically added the following to portion to subsection “Neuronal circuit models entail P1 neurons activating NPF^M^ neurons”: “While NPF^M^, P1 and NPFR neurons are essential for regulating courtship by responding to prior mating experience, and may do so through a recurrent inhibitory loop (Figure 8F), our data do not exclude other models. Part of the argument in favor of the recurrent inhibitory loop model is that the GRASP analysis suggests that NPFR neurons make direct connections with P1 neurons. Moreover, by coupling chemogenetic manipulation and Ca^2+^ imaging, we found that activation of NPFR neurons activate P1 neurons. However, NPFR neurons are widely distributed, and our data do not resolve whether the NPFR neurons that activate P1 neurons are the same subset of NPFR neurons that are the direct downstream target of NPF^M^ neurons. Thus, one alternative to the recurrent inhibitory motif is a feedforward parallel model, in which target neurons (X neurons) control courtship drive by receiving parallel input from P1 neurons and NPFR neurons (Figure 8G). This latter model posits that P1 neurons activate X neurons, and at the same time, send axonal branches to activate NPF^M^ neurons, which then act through NPFR neurons, and suppress the target neurons through a feedforward mechanism. Future experiments that resolve the anatomical and functional diversity of NPFR neurons should distinguish between the recurrent inhibitory versus feedforward parallel model, which ensure proper courtship output in accordance with a male’s internal drive state.”

3. Source data for behavioral and histology experiments.3A) Raw data for behavioral experiments should be included. This is especially important for the issue regarding the data in Figure 4B raised in the major comments. Without the raw data, it is impossible to statistically re-analyze the data. Perhaps it would also be useful to show statistical values relevant for non-normal distributions (alternatives to the mean, standard deviation and standard error), given that the statistics used involve the non-parametric Mann-Whitney U test, which concerns median.

All of the raw data and statistical values for the behavioral experiments are now provided in MS Word and Excel source data files.

3B) Also, GRASP experiments (Figure 5A-C, G-I, Figure 8A, B) are not informative unless (1) examples of negative controls, most importantly without a driver for GFP11 fragments), and (2) description of z-stack, are present. Authors use female brains as a proxy for negative controls, but it is not a fair comparison given that the expression of GFP1-10 (under the control of R71G01-GAL4) are likely different between the two sexes. Also, it is not clear whether the presented whole brain images are z-stack across the entire brain or a partial z-stack. The full z-stack is necessary to address whether the GRASP signal they presented in the panels above are the only signals they observed across the brain.

When we conducted our experiments, we examined multiple brains in which there is no driver for the *spGFP11*. These include a representative image of a *UAS-spGFP110, LexAop-spGFP11/NP21-Gal4* male brain, which is now shown in Figure 5C. In addition, we included a representative *UAS-spGFP1-10, LexAop-spGFP11/R71G01Gal4* male brain in Figure 5I. This latter control also applies to Figure 8A and B. To provide space for these panels in Figure 5C and I, we removed former Figure 5B and H, which were zoomed in regions of Figure 5A and G. Full brain stacks and videos of image sequences for the experimental samples and negative controls are included in the source data file. While there is some sparse, weak signals in controls without a driver for *spGFP11*, we do not detect signals in the LPC structure in any of the control brains, which is the key region that we find GRASP signals. We have also provided descriptions of the z-stacks in the legends to Figure 5 and Figure 8. Z projections of full stacks as well as videos showing image sequences across entire brains in the source data file.